# Opposing roles for DNA replication initiator proteins ORC1 and CDC6 in control of Cyclin E gene transcription

Manzar Hossain, Bruce Stillman*

Cold Spring Harbor Laboratory, Cold Spring Harbor, United States

**Abstract** Newly born cells either continue to proliferate or exit the cell division cycle. This decision involves delaying expression of Cyclin E that promotes DNA replication. ORC1, the Origin Recognition Complex (ORC) large subunit, is inherited into newly born cells after it binds to condensing chromosomes during the preceding mitosis. We demonstrate that ORC1 represses Cyclin E gene (CCNE1) transcription, an E2F1 activated gene that is also repressed by the Retinoblastoma (RB) protein. ORC1 binds to RB, the histone methyltransferase SUV39H1 and to its repressive histone H3K9me3 mark. ORC1 cooperates with SUV39H1 and RB protein to repress E2F1-dependent CCNE1 transcription. In contrast, the ORC1-related replication protein CDC6 binds Cyclin E-CDK2 kinase and in a feedback loop removes RB from ORC1, thereby hyper-activating CCNE1 transcription. The opposing effects of ORC1 and CDC6 in controlling the level of Cyclin E ensures genome stability and a mechanism for linking directly DNA replication and cell division commitment.

*For correspondence: stillman@cshl.edu

**Competing interests:** The authors declare that no competing interests exist.

## Introduction

In addition to its role in the initiation of DNA replication, ORC1, the largest subunit of the Origin Recognition Complex (ORC) controls Cyclin E-dependent duplication of centrosomes and centrioles in cells by acting as an inhibitor of Cyclin E-CDK2 activity (*Hemerly et al., 2009*; *Hossain and Stillman, 2012*). The Cyclin E-CDK2 kinase inhibitory activity is compromised by *ORC1* Meier-Gorlin syndrome mutations that also alter the interaction between ORC1 and histone H4K20me2 (*Hossain and Stillman, 2012*; *Kuo et al., 2012*; *Zhang et al., 2015*; *Bicknell et al., 2011b*; *Bicknell et al., 2011a*; *de Munnik et al., 2012*). Unlike the well-characterized yeast complex that is a stable, six subunit complex throughout the cell division cycle, ORC in human cells is a very dynamic complex (*Sasaki and Gilbert, 2007*; *DePamphilis, 2005*). ORC1 binds to mitotic chromosomes as cells enter into mitosis (*Kara et al., 2015*; *Okuno et al., 2001*), and in human cells, it is modified by ubiquitin and then degraded during the G1 to S phase transition (*Abdurashidova et al., 2003*; *Kara et al., 2015*; *Kreitz et al., 2001*; *Mendez et al., 2002*; *Ohta et al., 2003*; *Siddiqui and Stillman, 2007*; *Tatsumi et al., 2000*). The assembly of the full ORC occurs in mid G1 phase of the cell division cycle in preparation for its role in assembly of the pre-replicative complex (pre-RC) at sites across chromosomes (*Kara et al., 2015*; *Siddiqui and Stillman, 2007*).

The ORC1-related protein CDC6 is also required for pre-RC assembly, but it is targeted for proteasome degradation by the SCF^Cyclin F ubiquitin ligase complex late in the cell cycle and the anaphase-promoting complex/cyclosome (APC/C) in early G1 phase and then stabilized in mid G1 phase by Cyclin E-CDK2-mediated phosphorylation (*Mailand and Diffley, 2005*; *Petersen et al., 2000*; *Walter et al., 2016*). This phosphorylation is mediated by the direct interaction between Cyclin E and CDC6 and CDC6 and Cyclin E-CDK2 cooperate to promote the initiation of DNA replication (*Coverley et al., 2002*; *Furstenthal et al., 2001*; *Cook et al., 2002*).

**eLife digest** Living cells must replicate their DNA before they divide so that the newly formed cells can each receive an identical copy of the genetic material. Before DNA replication can begin, a number of proteins must come together to form so-called pre-replicative complexes at many locations along the DNA molecules. These protein complexes then serve as landing pads for many other DNA replication proteins.

One component of the pre-replicative complex, a protein called ORC1, helps to recruit another protein called CDC6 that in turn acts with Cyclin E to promote the replication of the DNA. Cyclin E is a protein that is only expressed when cells commit to divide. Previous research has shown that a lack of ORC1 causes the levels of Cyclin E to rise in human cells, but it was not understood how cells regulate the levels of Cyclin E.

Now, Hossain and Stillman show that the ORC1 protein switches off the gene that encodes Cyclin E early on in newly born cells, and therefore prevents the Cyclin E protein from being produced. The experiments show that ORC1 does this by binding near one end of the gene for Cyclin E and interacting with two other proteins to inactivate the gene. Thus, ORC1 establishes a period when Cyclin E is absent from a newly formed cell. This essentially gives the cell time to 'decide' (based on external cues and its own signaling) whether it will divide again or enter into a non-dividing state.

When a cell does decide to divide, the levels of CDC6 rise. CDC6 is another component of the pre-replicative complex and Hossain and Stillman find that CDC6 works to counteract the effects of ORC1 and reactivate the gene for Cyclin E. This activity leads to a dramatic increase in the production of Cyclin E, which in turn allows the cells to commit to another round of DNA replication and division.

The opposing effects of ORC1 and CDC6 control the levels of Cyclin E and provide a link between DNA replication and a cell's decision to divide. Further work is now needed to see whether ORC1 inactivates other genes in addition to the one that encodes Cyclin E.

As proliferating cells divide, they must make a decision whether to continue to proliferate or enter into proliferative quiescence. This decision is made by a complex regulatory process known as START in yeast and the restriction point in mammalian cells (*Johnson and Skotheim, 2013*). Key among these regulators are the Cyclin D-CDK4/6 kinases that mono-phosphorylate the retinoblastoma (RB) protein and contributes to the release of repression of E2F-transcription factors (*Narasimha et al., 2014*; *Ewen et al., 1993*; *Hinds et al., 1992*; *Lundberg and Weinberg, 1998*; *Resnitzky et al., 1994*). E2F1-regulated genes include genes encoding Cyclin E (*CCNE1*) and CDC6 (*Hateboer et al., 1998*; *Ohtani et al., 1998*; *Yan et al., 1998*; *DeGregori et al., 1995*). Cyclin E-CDK2 amplifies the phosphorylation of RB, but how this is achieved is not known (*Narasimha et al., 2014*). Here, we demonstrate that ORC1 is required for repression of the gene encoding Cyclin E and that CDC6 is involved in relieving the repression in cooperation with Cyclin E-CDK2. We suggest that ORC1 establishes a period in the newly born cells during which Cyclin E is not expressed, allowing time for the cells to decide whether to proliferate again or enter into replicative quiescence.

## Results

### ORC1 binds to RB and SUV39H1 and represses *CCNE1* gene transcription

Apart from its role in DNA replication, human ORC1 controls centriole and centrosome copy number by binding and inhibiting the kinase activity of Cyclin E-CDK2 (*Hemerly et al., 2009*). That work suggested that ORC1 might also control Cyclin E by regulating its protein level during the G1 phase of the cell division cycle. To determine if this was the case, ORC1 was depleted using siRNA in U2OS cells that had been synchronized in mitosis by nocodazole treatment and released into the next cell cycle. As early as 9 hr post release, Cyclin E protein level was elevated in ORC1-depleted U2OS cells, compared to control siRNA treated cells (*Figure 1A*). The expression of *CCNE1* mRNA

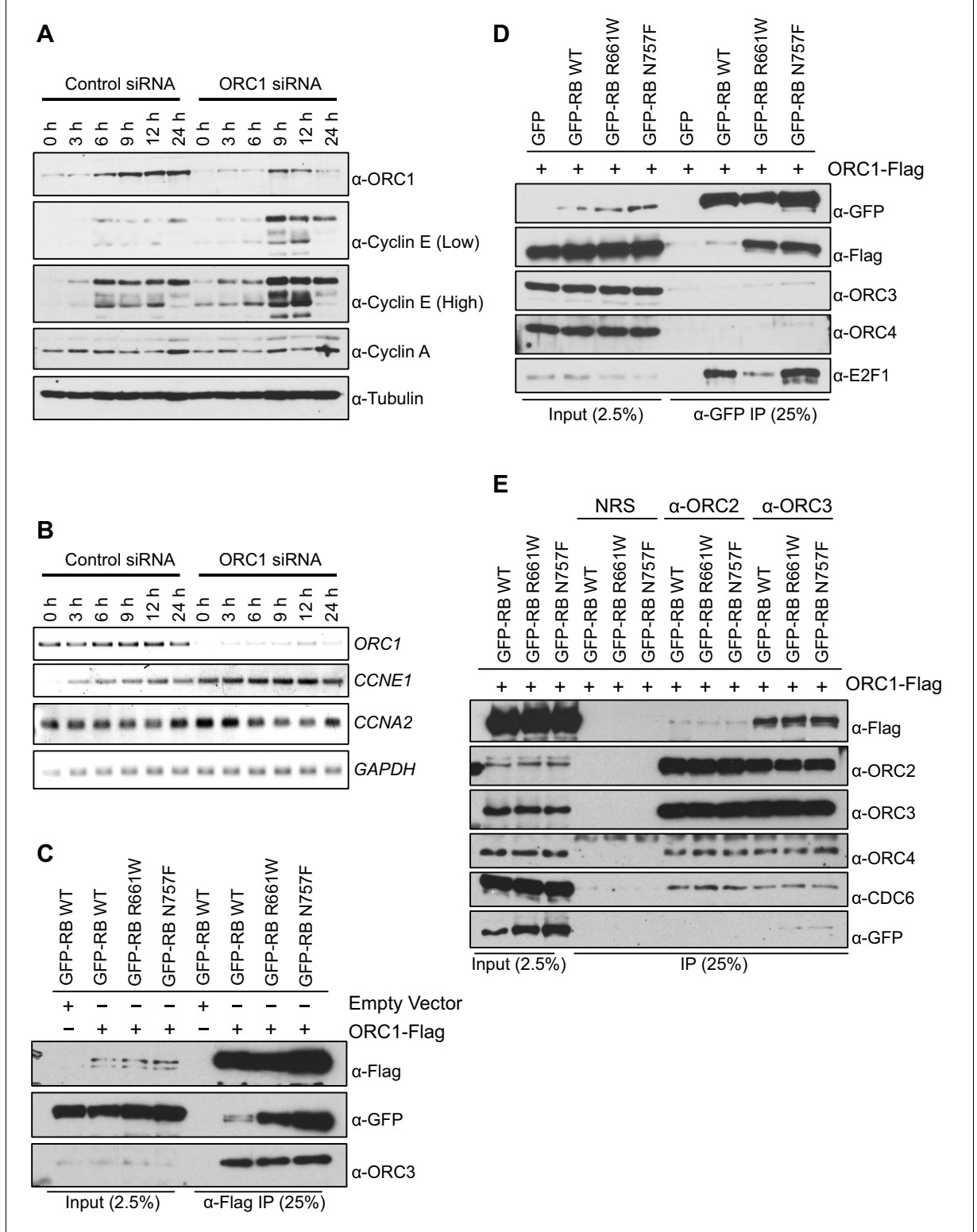

**Figure 1.** ORC1 represses Cyclin E gene expression and interacts with RB. (**A–B**) Nocodazole arrested U2OS cells were transfected with control or ORC1 siRNA then released into the next cycle. (**A**) protein levels were estimated by immunoblotting with antibodies against ORC1, Cyclin E, Cyclin A and α-Tubulin. Low and high indicates different exposures. (**B**) mRNA levels of *ORC1*, Cyclin E (*CCNE1),* Cyclin A (*CCNA2*) and *GAPDH.* Quantitation of mRNA levels from multiple experiments is shown in *Figure 1—figure supplement 1*. (**C–E**) Interaction between ORC1 and RB or its pocket mutants. GFP, GFP-tagged wild-type or mutant RB were co-transfected in HEK293 cells with either ORC1-Flag or empty vector. Immunoprecipitation with anti-
*Figure 1 continued on next page*

Figure 1 continued

Flag antibody (C) or GFP antibody (D) from cell lysates followed by immunoblotting with the indicated antibodies. (E) Cell lysate from HEK293 cells overexpressing GFP-tagged wild type or mutant RB and ORC1-Flag were immunoprecipitated with normal rabbit serum (NRS) or ORC2 or ORC3 antibodies, immunoblotted with the indicated antibodies. Binding of ORC1 to wild-type and RB mutants and the effect of Cyclin E-CDK2 is shown in *Figure 1—figure supplement 2*. GFP, Green fluorescent protein; ORC, Origin Recognition Complex; RB, Retinoblastoma.

The following figure supplements are available for figure 1:

**Figure supplement 1.** ORC1 represses Cyclin E gene expression and interacts with RB.

**Figure supplement 2.** ORC1 represses Cyclin E gene expression and interacts with RB.

increased at all times following ORC1 depletion in synchronized U2OS cells (*Figure 1B*) and quantitation of multiple experiments showed significant increases from 6–24 hr post release (*Figure 1—figure supplement 1A–D*). This data suggests that ORC1 inhibits *CCNE1* gene expression.

Transcription of the gene encoding Cyclin E (*CCNE1*) is known to be regulated by the E2F1 transcription factor and to be repressed by RB protein (*Nielsen et al., 2001*; *Geng et al., 1996*; *DeGregori et al., 1995*), a member of the so-called 'pocket protein' family (*Dick and Rubin, 2013*; *Dyson, 1998*; *Giacinti and Giordano, 2006*). Since ORC1 binds RB (*Mendoza-Maldonado et al., 2010*), we explored a role for ORC1 in transcriptional repression of the *CCNE1* gene.

RB was expressed as a fusion to the maltose binding protein (MBP) and it bound to $S^{35}$-labelled ORC1 protein (*Figure 1—figure supplement 2A*). RB binds other binding partners dependent on a canonical LxCxE motif and although ORC1 has a conserved $LPCR^D/_E$ sequence, it was not required for the interaction between ORC1 and RB (*Figure 1—figure supplement 2A and B*). Consistent with this finding, two pocket mutants within RB (R661W and N757F) that are defective in binding to LxCxE containing proteins (*Chen and Wang, 2000*) showed no defect in binding to ORC1 in vitro (*Figure 1—figure supplement 2C*) or in vivo (*Figure 1C*). In fact, the mutant RB proteins bound ORC1 better than the wild type, perhaps due to loss of competition between ORC1 and other RB binding proteins or due to conformational changes in RB. When Green Fluorescent Protein (GFP)-RB fusion protein was expressed in cells with ORC1-Flag-tagged protein, the ORC1-Flag protein bound WT and pocket mutant RB, but other ORC subunits did not bind to RB (*Figure 1D*). We confirmed this observation by showing that immunoprecipitation with anti-ORC2 or anti-ORC3 antibodies failed to precipitate GFP-RB or its mutants, but did bind ORC1-Flag (*Figure 1E*). Given that the ORC1-RB interaction is independent of other ORC subunits, it suggests that ORC1 has additional functions that are separate from its role in DNA replication.

The phosphorylation of RB is an important step that relieves repression of E2F target genes (*Rubin, 2013*). In the presence of purified Cyclin E-CDK2 kinase plus ATP RB no longer bound to ORC1 (*Figure 1—figure supplement 2A and C*), suggesting that Cyclin E might feedback and relieve repression of the *CCNE1* gene by disrupting the RB-ORC1 interaction.

RB interacts with chromatin and histone-modifying enzymes to repress E2F1 transcription activity (*Dick and Rubin, 2013*). Specifically, it has been suggested that RB binds to the SUV39H1 histone methyltransferase that tri-methylates histone H3K9 and then HP1α binds to histone H3K9me3 and that these direct interactions contribute to repression of *CCNE1* transcription (*Nielsen et al., 2001*). It is important to note, however, that others suggested that another protein mediates the RB-SUV39H1 interaction (*Vandel et al., 2001*). Since ORC1 interacts with both RB and HP1α (*Prasanth et al., 2010*), we tested whether ORC1 also interacts with SUV39H1. Using purified proteins we found that both RB and ORC1 directly bound to SUV39H1, and ORC1 directly bound to CDC6 (*Saha et al., 1998*) (*Figure 2A* and *Figure 2—figure supplement 1*). Under these conditions, ORC1 bound to HP1α weakly, but the binding increased when higher levels of GST-HP1α were added, consistent with published data (*Prasanth et al., 2010*). Although both RB and ORC1 directly interacted with SUV39H1, ORC1 bound SUV39H1 much better than RB (*Figure 2—figure supplement 2A–D*). Consistent with the in vitro data, we observed that anti-SUV39H1 antibodies precipitated ORC1 and RB in the RB-positive cells (U2OS and MCF7), with ORC1 being the predominant interacting protein in MCF7 cells (*Figure 2B*). Furthermore, the interaction between SUV39H1 with ORC1 protein did not require RB because anti-ORC1 antibodies co-precipitated SUV39H1 from RB-

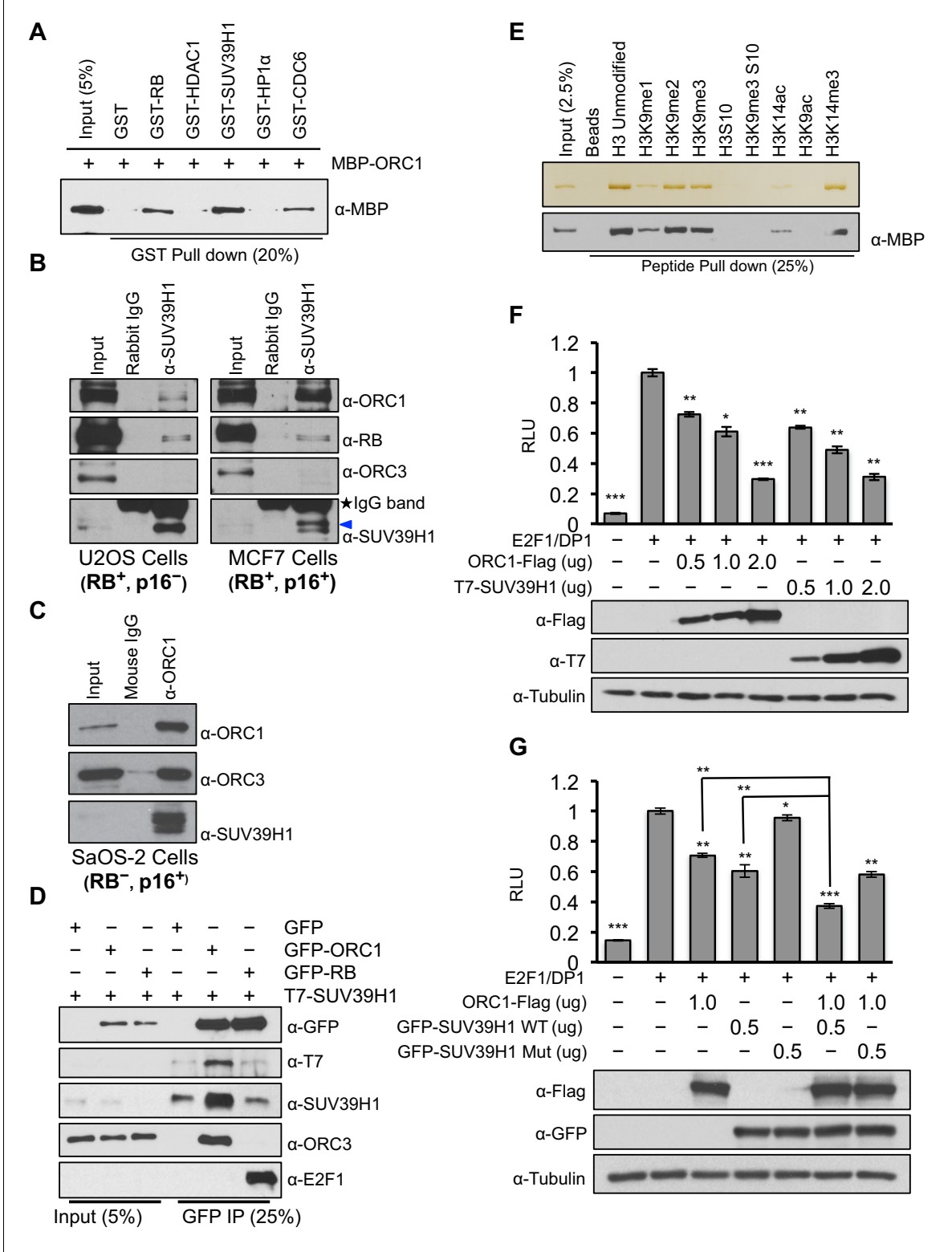

**Figure 2.** ORC1 binds SUV39H1 to control Cyclin E gene transcription. (**A**) Purified MBP-ORC1 and various GST-fused proteins were mixed and proteins bound in a GST-pull down were detected by immunoblotting with anti-MBP antibodies. The purified proteins are shown in *Figure 2—figure supplement 1*. (**B**) U2OS and MCF7 cell lysates were immunoprecipitated with SUV39H1 antibody and immunoblotted with the indicated antibodies. Rabbit IgG served as control antibody. Asterisk indicates the cross-reacting antibody band; arrow indicates the SUV39H1 protein. (**C**) Immunoprecipitation from RB-negative SaOS-2 cell lysates with ORC1 antibody or IgG and immunoblotted with antibodies against ORC1, SUV39H1 or

*Figure 2 continued on next page*

Figure 2 continued

ORC3. (D) HEK293 cells were transiently co-transfected with GFP, GFP-ORC1 or GFP-RB plus T7-SUV39H1 plasmids (2.5 μg each). GFP antibody immunoprecipitates were immunoblotted with the indicated antibodies. The interaction between ORC1 and SUV39H1 and between RB and SUV39H1 is shown with purified proteins and quantitated in *Figure 2—figure supplement 2*. Higher levels of RB are required to demonstrate an interaction with SUV39H1 in vivo and ORC1 interacts with the SET domain of SUV39H1, *Figure 2—figure supplement 3*. (E) MBP-ORC1 was incubated with bead-bound histone peptides with or without the indicated modifications and bound MBP-ORC1 was observed by immunoblotting with anti-MBP antibody (lower box) or silver staining (upper box). (F–G) Wild-type *CCNE1*-luciferase reporter assay in U2OS cells. U2OS cells were transiently co-transfected with 500 ng of 10–4 *CCNE1* promoter, 50 ng E2F1, 50 ng DP1 and 20 ng pCMV-LacZ plasmids along with the indicated amounts ORC1 and/or SUV39H1 plasmids. (F) Increasing amounts of ORC1-Flag or T7-SUV39H1 repress Cyclin E gene promoter. Experiments were carried out in triplicate. Expression of proteins was confirmed by Immunoblot; α-Tubulin as loading control. Statistical analysis was performed using the Student's t test. *$p<0.05$; **$p<0.005$; ***$p<0.001$. (G) ORC1-Flag cooperates with wild type but not mutant SUV39H1 to repress *CCNE1* gene expression. The experiments were carried out in triplicate. Expression of proteins was confirmed by Western blots. α-Tubulin served as a control for equal loading of each sample. Statistical analysis was performed using the Student's t test. *$p<0.01$; **$p<0.005$; ***$p<0.0001$. Repression of transcription by ORC1 and SUV39H1 was also demonstrated using an artificial promoter and tethering the proteins via the GAL4 DNA binding domain in *Figure 2—figure supplement 4*. ORC, Origin Recognition Complex; MBP, Maltose binding protein; GST, Glutathione S transferase.

The following figure supplements are available for figure 2:

**Figure supplement 1.** ORC1 binds SUV39H1 to control Cyclin E gene transcription.

**Figure supplement 2.** ORC1 binds SUV39H1 to control Cyclin E gene transcription.

**Figure supplement 3.** ORC1 binds SUV39H1 to control Cyclin E gene transcription.

**Figure supplement 4.** ORC1 binds SUV39H1 to control Cyclin E gene transcription.

negative SaOS-2 cells (*Figure 2C*). This observation was confirmed when GFP-RB or GFP-ORC1 were transiently expressed with T7-SUV39H1 in 293 cells; ORC1 more readily bound to SUV39H1 (*Figure 2D*), whereas higher levels of GFP-RB were required to observe an interaction with SUV39H1 (*Figure 2—figure supplement 3A*). Domain mapping demonstrated that SUV39H1 interacted with ORC1 through its SET domain containing C-terminus, which is required for its histone methyltransferase (HMT) activity. Moreover, SUV39H1 interacted with ORC1 and ORC1 mutants that could not be phosphorylated by Cyclin-CDK and did not interact in a complex with other ORC subunits, similar to the ORC1-RB interaction (*Figure 2—figure supplement 3B–C*). These data suggest that the ORC1-SUV39H1 interaction does not require other ORC subunits or CDK phosphorylation of ORC1, and therefore ORC1 plays a role independent of its role in DNA replication.

Since SUV39H1 tri-methylates histone H3K9 after its de-acetylation by HDAC1, a prerequisite step for establishing *CCNE1* gene repression, we explored the ORC1 protein interactions with different histone H3 modifications (*Nicolas et al., 2003*; *Stewart et al., 2005*; *Vaute et al., 2002*). Purified ORC1 bound to unmodified or mono-, di- or tri-methylated H3K9 or H3K14-tri-methyl modifications, while the interaction was abolished when histone H3 was acetylated at the same positions or when H3 serine-10 was phosphorylated, which normally occurs during mitosis (*Figure 2E*). Thus, ORC1 can bind to RB, SUV39H1 and to the repressive H3K9-me3 modification on histone H3, suggesting that it might mediate repression of E2F1-dependent *CCNE1* transcription.

To test if ORC1 has a role in repression of the E2F1-regulated *CCNE1* gene, the *CCNE1* promoter was linked to the luciferase-coding region to create a reporter for *CCNE1* gene transcription (*Geng et al., 1996*). Transfection of plasmids expressing E2F1 and its binding partner DP1 activated gene expression, whereas expression of ORC1 or SUV39H1 protein repressed *CCNE1* expression in a dose-dependent manner (*Figure 2F*). In a GAL4-based reporter gene assay, expression of either GAL4-ORC1 or GAL4-SUV39H1 also repressed transcription of the reporter gene in a dose-dependent manner (*Figure 2—figure supplement 4A*). In contrast, neither expression of ORC3 nor ORC4 altered E2F1-driven *CCNE1* transcription (*Figure 2—figure supplement 4B*). Moreover, SUV39H1 co-operated with ORC1 to further repress the *CCNE1* promoter, but a mutant of SUV39H1 (H324K) (*Li et al., 2002*; *Rea et al., 2000*; *Stewart et al., 2005*) that has lost its catalytic activity was unable to do so (*Figure 2G*). We therefore conclude that ORC1-SUV39H1 co-operation for transcriptional repression of the *CCNE1* gene is mediated through the HMT activity of SUV39H1.

## Cell-cycle-dependent association of ORC1, RB, SUV39H1 and CDC6 proteins with the *CCNE1* gene promoter

Having established that ORC1 can repress *CCNE1* gene transcription in cooperation with the methyltransferase activity of SUV39H1, we investigated the positioning of ORC1, RB and SUV39H1 proteins within the *CCNE1* promoter in vivo. A previous publication (*Dellino et al., 2013*) reported ORC1 chromatin immuno-precipitation following crosslinking (ChIP) and re-analysis this whole genome Chip-Seq data revealed a weak ORC1 peak (peak height of 7 reads) within the *CCNE1* promoter; however, a duplicate was not reported. Therefore, to test whether ORC1 associated with the *CCNE1* promoter, we modified a chromatin immunoprecipitation (ChIP) method for ORC1 initially using asynchronously growing MCF7 cells and the immunoprecipitated DNA was analyzed by polymerase chain reaction (PCR) using multiple primer pairs across the *CCNE1* promoter (*Figure 3A*). ORC1 bound to the *CCNE1* promoter encompassing the region from -280 to +63 base pairs (probes E and F), a region that is known to bind the E2F1 transcription factor (see Gene Expression Omnibus [GEO] transcription factor binding site accession numbers GSM935484 and GSM935477) and contains five E2F1 consensus binding sites (red bars, *Figure 3A*). Our ChIP analysis also showed that RB bound to the same regions bound by ORC1, but not to other probes (*Figure 3B*). When a similar ChIP analysis was performed using antibodies targeted to SUV39H1, histone H3K9me3 and CDC6, these proteins bound to the same region (-342 to +63) of the *CCNE1* promoter (*Figure 3C*, also see *Figure 4* below).

We next studied the temporal dynamics of protein binding to the CCNE1 promoter during the G1 phase of the cell cycle in synchronized U2OS cells. In this analysis, cells were blocked in mitosis with nocodazole and released into the next G1 phase and ChIP analyses for ORC1, CDC6, RB and SUV39H1 were performed using primer pairs B and E (*Figures 4A and B*). ORC1 and RB bound to the probe E region, but not the probe B region, at 3 hr post mitosis, and then binding was reduced at 6 hr and eliminated by 9 hr (*Figure 4A*), even though ORC1 protein remained in the cell (*Figure 4C*). SUV39H1 was detected at 3 and 6 hr, but not at 9 hr. Interestingly, CDC6 transiently bound to the promoter only at 6 hr (*Figure 4B*), precisely the time when Cyclin E proteins levels dramatically increase (*Figure 4C*). In the following section, we investigated the physiological role of CDC6 binding to *CCNE1* promoter.

## CDC6 cooperates with Cyclin E-CDK2 to remove RB from ORC1 and activate *CCNE1* gene transcription

Cyclin E-CDK2 cooperates with CDC6 to stimulate entry from G1 phase into S-phase of the mammalian cell division cycle (*Cook et al., 2002*; *Coverley et al., 2002*; *Hateboer et al., 1998*). Since CDC6 protein levels increase during late G1 phase (*Hateboer et al., 1998*; *Mendez and Stillman, 2000*) and CDC6 binds ORC1 (*Figure 2A*), we hypothesized that CDC6 may help alleviate ORC1-mediated repression of the *CCNE1* gene. CDC6 and Cyclin E-CDK2 were purified and increasing amounts were titrated into a mixture containing purified RB and ORC1 (*Figure 5—figure supplement 1*) and the interaction between RB and ORC1 was monitored. MBP-ORC1 bound to CDC6 (*Figure 5A*) and increasing amounts of Cyclin E-CDK2 did not interfere with this interaction (*Figure 5B*). Moreover, increased binding between CDC6 and ORC1 did not interfere with binding between ORC1 and RB (*Figure 5B*). Cyclin E-CDK2 disrupted the interaction between ORC1 and RB (*Figure 5C* and *Figure 2—figure supplement 2A*), but it took relatively high levels of Cyclin E-CDK2. Importantly, the addition of CDC6 along with CyclinE-CDK2 cooperatively disrupted the interaction between ORC1 and RB, even at low concentrations of Cyclin E-CDK2 (*Figure 5C*). As shown before with *Xenopus* proteins (*Furstenthal et al., 2001*), human CDC6 bound to Cyclin E-CDK2 in a manner dependent on the CDC6 Cyclin 'Cy' binding motif CDC6[94]RRL[96] but not to the CDC6[94]ARA[96]'Cy' mutant (*Figure 5D*).

Based upon the biochemical data, we hypothesized that CDC6 and Cyclin E-CDK2 cooperated to relieve ORC1-SUV39H1-RB-mediated repression of *CCNE1*. If this is the case, then CDC6 overexpression should increase *CCNE1* expression in vivo. To test this hypothesis, we expressed GFP-tagged CDC6 wild type or its 'Cy' mutant in nocodazole arrested U2OS cells and released them from the nocodazole block to estimate the level of endogenous Cyclin E protein at different times in G1. Expression of wild type CDC6 but not its 'Cy' mutant led to an increase in the endogenous level of Cyclin E protein (*Figure 5E*). In fact, the Cdc6 'Cy' mutant acted as a dominant negative to

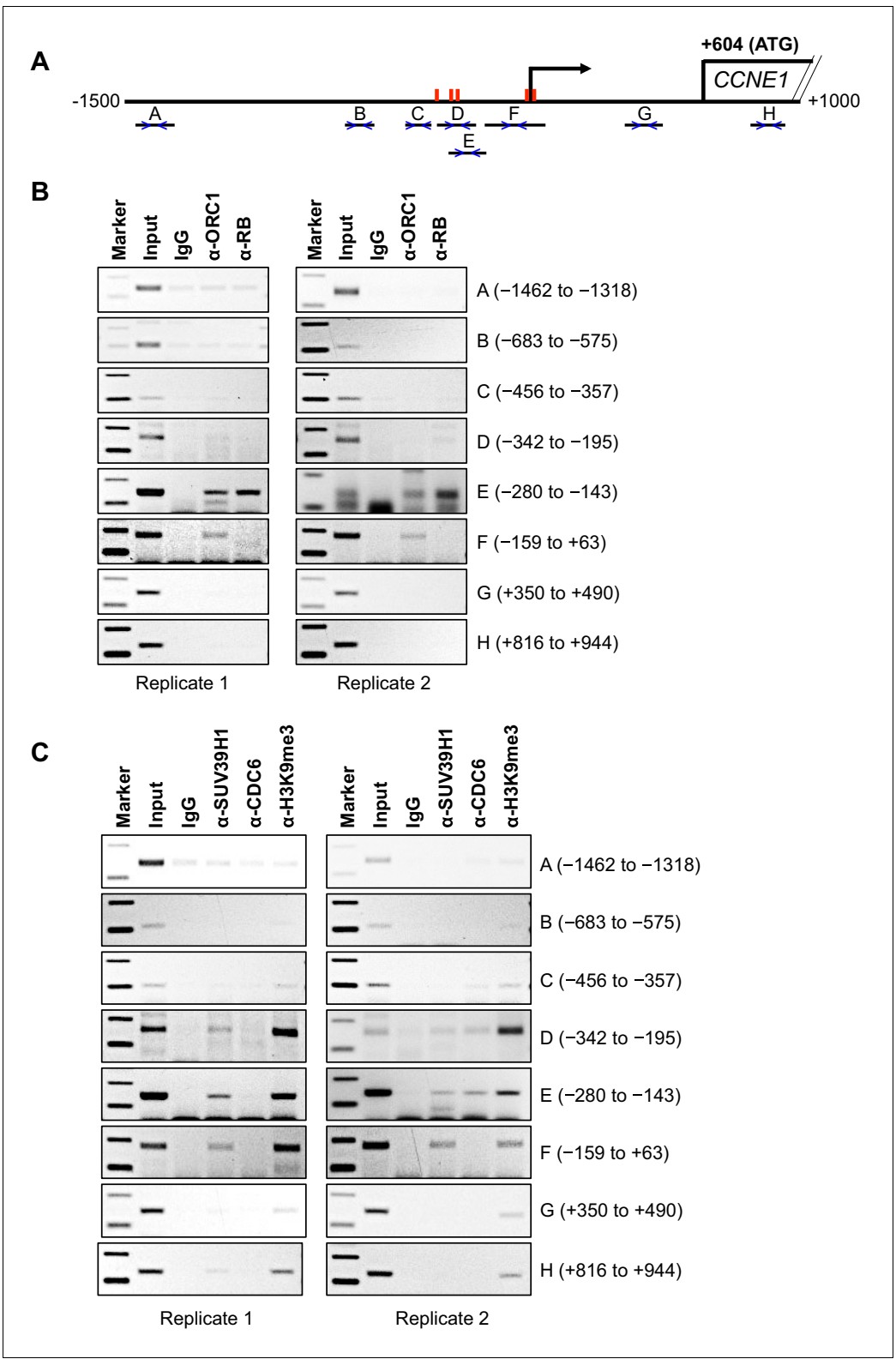

**Figure 3.** Binding of ORC1, RB, SUV39H1 and CDC6 proteins to the *CCNE1* promoter. (**A**) Schematic of the *CCNE1* promoter and the regions amplified with different primer pairs used for ChIP assay were indicated as follows: A (−1462 to −1318); B (−683 to −575); C (−456 to −357); D (−342 to −195); E (−280 to −143); F (−159 to +63); G (+350 to +490); H (+816 to +944). The red bars indicate five E2F1 consensus sites. The truncated box indicates the first exon of the *CCNE1* gene. (**B–C**) The occupancy of ORC1, RB, SUV39H1 and CDC6 proteins was

*Figure 3 continued on next page*

*Figure 3 continued*

analyzed by chromatin immunoprecipitation at the *CCNE1* promoter in asynchronous growing MCF7 cells. ORC1 and RB are mouse antibodies, while SUV39H1 and CDC6 are rabbit antibodies. In the marker lanes, the two bands are 100 and 200 base pairs. The experiments were done in triplicate and two of these experiments are shown.

prevent normal Cyclin E expression. Consistent with these results, overexpression of CDC6 wild type but not its 'Cy' mutant further enhanced E2F1-DP1 activated transcription from the *CCNE1* promoter (*Figure 5F*). Based on these data, we suggest that CDC6 co-operates with Cyclin E-CDK2 kinase to abolish the interaction between RB and ORC1, contributing to alleviating the repression imposed by RB on the *CCNE1* gene, leading to *CCNE1* gene transcription.

Data presented so far suggests that ORC1-mediated *CCNE1* transcriptional repression also requires SUV39H1. We therefore transfected into RB⁺ U2OS cells a *CCNE1*-luciferase reporter plasmid with E2F1-DP1 and depleted ORC1, SUV39H1 or CDC6 using siRNA (*Figure 6A*). Depletion of

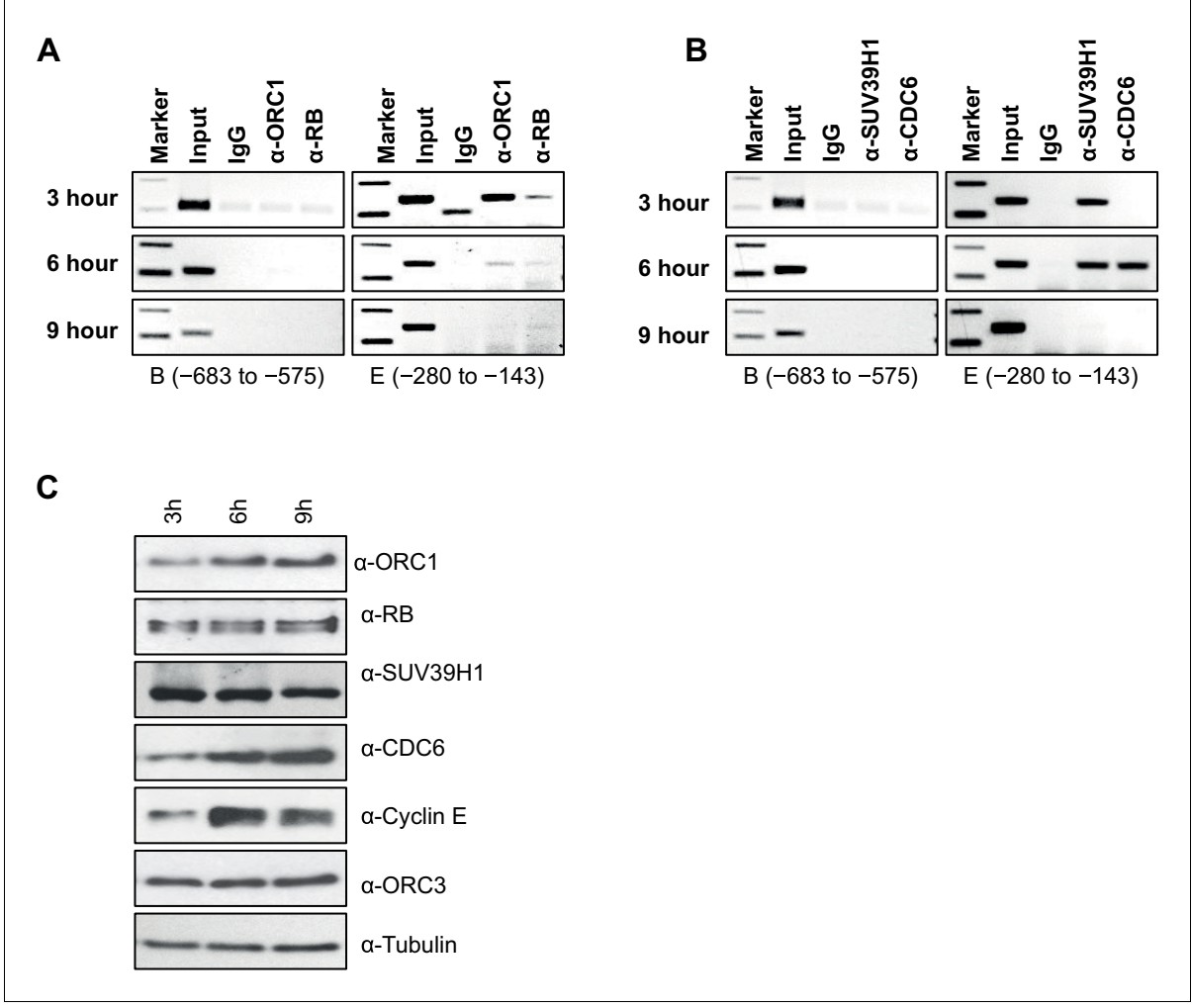

**Figure 4.** Dynamic association of ORC1, RB, SUV39H1 and CDC6 proteins to the *CCNE1* promoter during the cell cycle. (A–B) Nocodazole arrested U2OS cells were released for different times (3, 6 and 9 hr) and analyzed for occupancy of ORC1, RB, SUV39H1 and CDC6 proteins at the *CCNE1* promoter by ChIP assay. The primer pairs used to analyze two different regions of the *CCNE1* promoter are indicated. The experiments were done in triplicate with results similar to those shown. (C) Whole cell protein levels of nocodazole arrested and released U2OS cells at different time points (as indicated in hours) by immunoblotting with antibodies against ORC1, RB, SUV39H1, CDC6, Cyclin E, ORC3 and α-Tubulin. ORC, Origin Recognition Complex; RB, Retinoblastoma.

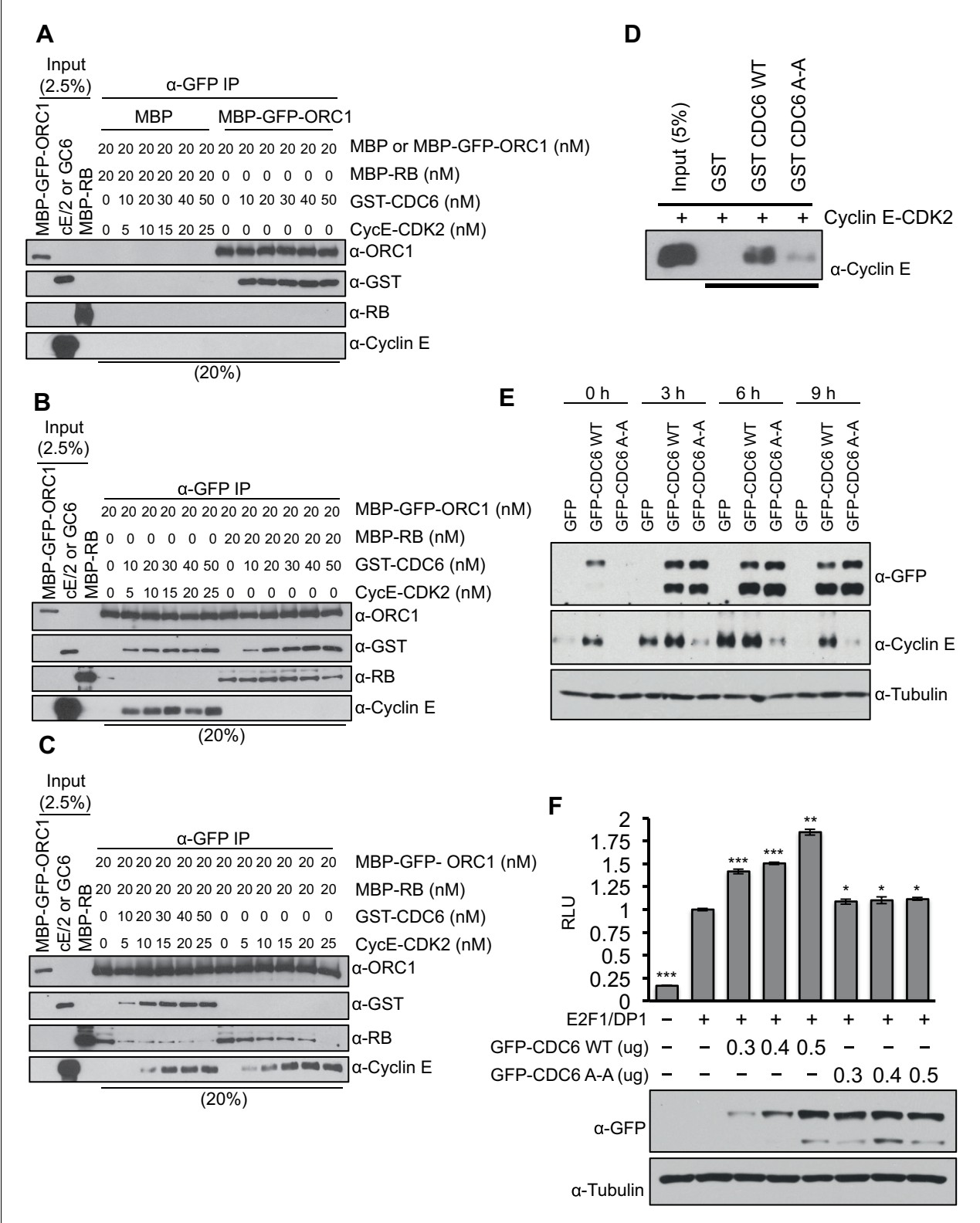

**Figure 5.** CDC6 co-operates with Cyclin E-CDK2 to activate E2F1-dependent *CCNE1* gene transcription. (A–C) Equimolar amounts of MBP-GFP-ORC1 and MBP-RB proteins were incubated with increasing amounts of GST-CDC6 and/or Cyclin E-CDK2. MBP-GFP-ORC1 protein was immunoprecipitated with GFP antibody, then immunoblotted with the indicated antibodies. The purified proteins used in these experiments are shown in *Figure 5—figure supplement 1* (A) MBP-GFP-ORC1 binds GST-CDC6. MBP protein served as control. (B) The binding of MBP-GFP-ORC1 protein to either GST-CCD6 in

*Figure 5 continued on next page*

*Figure 5 continued*

the presence of Cyclin E-CDK2 (left section) or MBP-RB (right section). (**C**) The binding of MBP-GFP-ORC1 to MBP-RB in the presence of increasing molar amounts of Cyclin E-CDK2 (right section) or both GST-CDC6 and Cyclin E-CDK2 (left section). (**D**) GST-pull down assay using GST-CDC6 wild type or CDC6$^{94}$ARA$^{96}$ mutant (CDC6A-A) with purified Cyclin E-CDK2 protein followed by Immunoblotting with Cyclin E antibody. GST protein served as control. (**E**) Nocodazole arrested U2OS cells were transfected with 500 ng of GFP, GFP-CDC6 wild type or CDC6$^{94}$ARA$^{96}$ mutant (CDC6A-A) plasmids, then released into the next cell cycle. At indicated times, whole cell extracts were immunoblotted with specific antibodies against GFP and Cyclin E. α-Tubulin served as loading control. (**F**), *CCNE1* promoter-luciferase reporter assay in U2OS cells. Cells transiently co-transfected with 500 ng of 10–4 *CCNE1* promoter, 50 ng E2F1, 50 ng DP1 and 20 ng pCMV-LacZ plasmids together with increasing amounts GFP-CDC6 WT or CDC6$^{94}$ARA$^{96}$ plasmids for 24 hr. Relative luciferase activity was normalized to co-transfected LacZ control. Experiments in triplicate. Protein expression determined by immunoblot; α-Tubulin as loading control. Statistical analysis was performed using the Student's t test. *$p<0.05$; **$p<0.001$; ***$p<0.0005$. GFP, Green fluorescent protein; MBP, Maltose binding protein; GST, Glutathione S transferase.

The following figure supplement is available for figure 5:

**Figure supplement 1.** CDC6 co-operates with Cyclin E-CDK2 to activate E2F1-dependent CCNE1 gene transcription.

either ORC1 or SUV39H1 with two different siRNAs led to significant increases in *CCNE1* promoter activity above its basal, E2F1/DP1-dependent level. In contrast, depletion of CDC6 protein had no effect on the basal, E2F1/DP1-dependent promoter activity (*Figure 6A*). We further investigated the binding to the *CCNE1* promoter of SUV39H1 and the presence of its product, the histone H3K9me3 mark, upon depletion of ORC1 protein in asynchronously growing U2OS cells by ChIP assay. Upon depletion of ORC1 compared to control siRNA the methyltransferase activity of SUV39H1 was drastically reduced on the *CCNE1* promoter as evident by a dramatic reduction in the histone H3K9m3 mark at the promoter, while the binding of SUV39H1 was only slightly reduced (*Figure 6B*). The reduced binding of H3K9me3 was most evident within the region -280 to -143 of the *CCNE1* promoter, where ORC1 binding was centered (*Figure 6B*, *Figure 6—figure supplement 1*). Depletion of ORC1 also resulted in reduced levels of the histone H3K9me3 mark as well as a slight reduction in the level of SUV39H1 protein (*Figure 6C*). Our results support the hypothesis that ORC1 is involved in transcription repression by recruiting the SUV39H1 protein, which thereby creates the histone H3K9me3 transcriptional repressor mark on *CCNE1* promoter.

To separate the function of ORC1 as a transcription co-repressor from its well-established role in DNA replication, we identified C-terminus truncation mutants (1–700 aa and 1–768 aa) of ORC1 (full length ORC1 is 1–861 aa) that were defective in binding to the other ORC subunits, but still capable of binding RB and SUV39H1 (*Figure 7A–C*). These ORC1 mutants were fully active in repressing *CCNE1* transcription (*Figure 7D*), demonstrating that the effects of ORC1 on *CCNE1* gene repression were not due to an indirect effect of the role of ORC in DNA replication.

## Discussion

ORC1 in plants is known to activate transcription in via a plant homeodomain (PHD) that is not found in animal and fungi ORC1 (*Sasaki and Gilbert, 2007*; *de la Paz Sanchez and Gutierrez, 2009*). In contrast, ORC1 in the budding yeast *S. cerevisiae* binds to the Sir1 protein and is involved in repression of transcription of mating type genes (*Bell et al., 1993*; *Triolo and Sternglanz, 1996*). In *Drosophila* and animal cells, ORC, including human ORC1, binds the HP1 proteins that are often involved in transcriptional repression, but a direct link between this interaction and transcription repression has not been established (*Auth et al., 2006*; *Badugu et al., 2003*; *Lidonnici et al., 2004*; *Pak et al., 1997*; *Prasanth et al., 2010*; *Sasaki and Gilbert, 2007*). In this report, we have established opposing roles for ORC1 and CDC6 in control of transcription of the Cyclin E gene *CCNE1*. ORC1, with RB and SUV39H1, bind to the *CCNE1* promoter adjacent to the E2F1 transcription factor binding sites and repress the E2F1-dependent promoter and this repression is relieved by CDC6 that is bound to Cyclin E-CDK2 kinase. Consistent with this model, CDC6 binds to the *CCNE1* promoter transiently, just at the time during G1 phase when Cyclin E levels increase dramatically. We suggest that low levels of both Cyclin E and CDC6, both E2F1-controlled genes, appear upon activation of Cyclin D-CDK4 (*Narasimha et al., 2014*). The expression of CDC6-Cyclin E-CDK2 provides positive feedback, disrupting the repressive RB-ORC1 interaction. The disruption of RB-ORC1

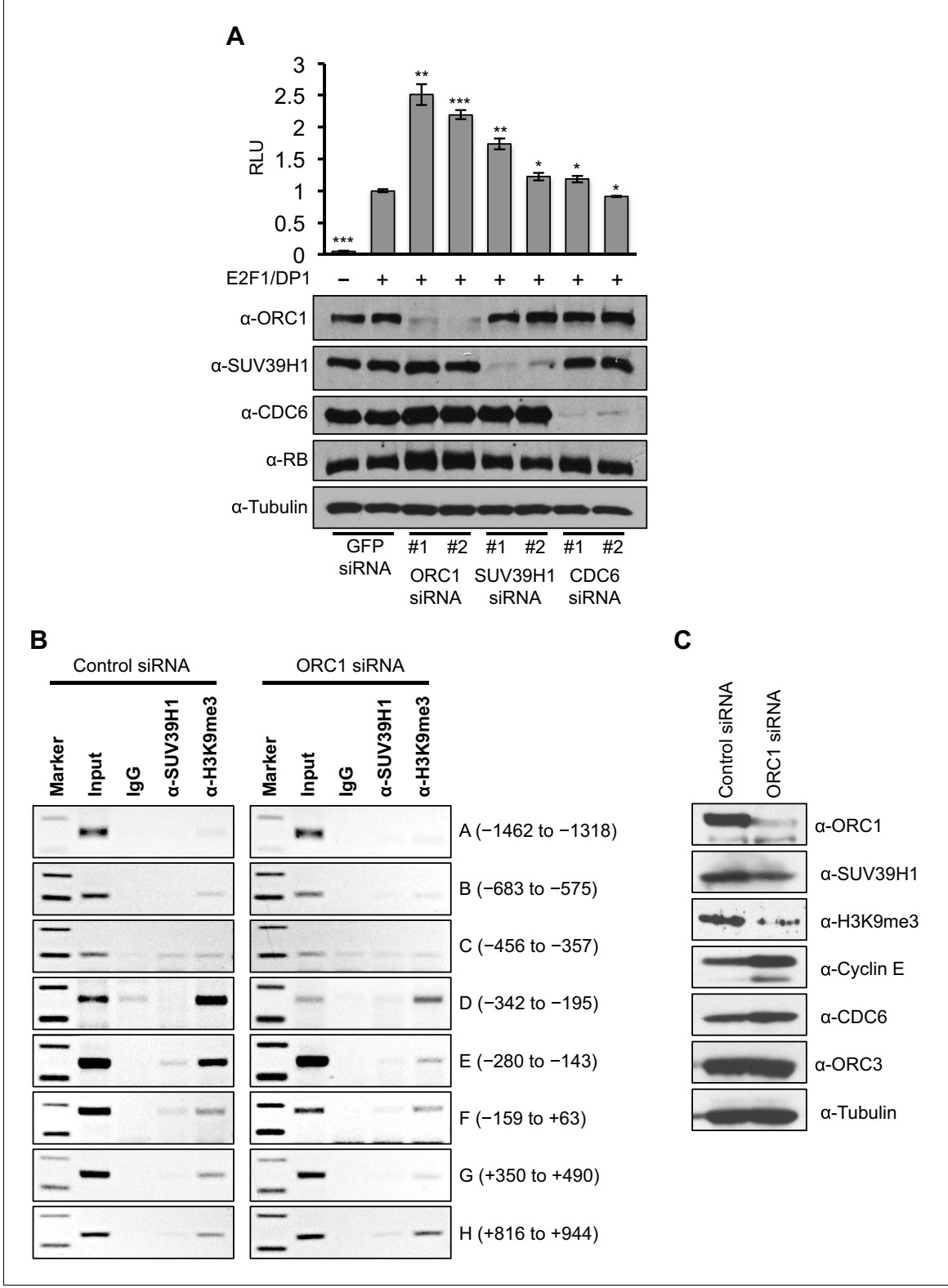

**Figure 6.** ORC1 depletion decreases association of SUV39H1 and H3K9me3 with the *CCNE1* promoter and increases *CCNE1* gene transcription. (**A**) *CCNE1*-luciferase reporter assay in U2OS cells. U2OS cells transiently co-transfected with 500 ng of 10–4 *CCNE1* promoter, 50 ng E2F1, 50 ng DP1 and 20 ng pCMV-LacZ. U2OS cells were also transiently transfected for 24 hr with two different siRNAs targeting either ORC1, SUV39H1 or CDC6. GFP siRNA was used as a control. Relative luciferase activity was normalized to co-transfected LacZ control. Depletion of proteins was confirmed by Immunoblot; α-Tubulin as loading control. Statistical analysis was performed using the Student's t test. *p<0.005; **p<0.001; ***p<0.0005. (**B**) The
*Figure 6 continued on next page*

*Figure 6 continued*

*CCNE1* promoter was analyzed for SUV39H1 binding and the presence of the H3K9me3 mark by ChIP assay in U2OS cells treated with either control siRNA or ORC1 siRNA for 48 hr. The experiments were done in triplicate and one experiment is shown. (**C**) Immunoblot of protein levels following control and ORC1 siRNA treatment at different times post nocodazole release. α-Tubulin was used as loading control.

The following figure supplement is available for figure 6:

**Figure supplement 1.** The ChIP qPCR bands were quantified using ImageJ software to analyze the extent of binding of SUV39H1 and histone H3K9me3 to the *CCNE1* promoter region (-280 to -143 bp) in ORC1 siRNA treated U2OS cells compared to control siRNA-treated cells.

interaction further weakens ORC1 association the with promoter and results in a dramatic reduction in the repressive histone H3K9me3 mark at the promoter, increasing *CCNE1* gene transcription and amplifying the levels of CDC6 and Cyclin E-CDK2 (*Figure 8*, bottom panel). The opposing expression levels of ORC1 and CDC6 during the cell cycle (*Figure 8*, top panel) are consistent with this model. The production of higher levels of Cyclin E-CDK2 and CDC6 can then function in pre-RC assembly, as they are known to do (*Cook et al., 2002*; *Coverley et al., 2002*; *Hateboer et al., 1998*).

Our work also shows that ORC1 binds to *CCNE1* promoter and recruits SUV39H1, forming the repressive H3K9me3 mark that can then bind the HP1 protein. Interestingly, HP1 is known to bind both SUV39H1 and ORC1 (*Prasanth et al., 2010*; *Stewart et al., 2005*). Thus, ORC1 participates in multiple protein-protein interactions to ensure stable repression of *CCNE1* and perhaps other E2F1-regulated genes during early G1 phase.

In this way ORC1, which binds mitotic chromosomes and is then inherited into the daughter cells (*Kara et al., 2015*; *Okuno et al., 2001*), creates an opportunity to repress *CCNE1* gene transcription and allow the newborn cells time to integrate information to decide whether to proliferate or exit the cell division cycle. If proliferation and cell division are destined to occur, Cyclin D-CDK4/6 mono-phosphorylates RB and primes it for Cyclin E-dependent activation of E2F regulated genes (*Narasimha et al., 2014*). We show that low levels of CDC6-Cycin-E-CDK2 can, in a feedback loop, amplify this commitment by antagonizing ORC1-RB interactions (*Figure 8*). Consistent with the model, we demonstrated that over-expression of CDC6, but not a CDC6 that cannot bind Cyclin E, enhances the levels of endogenous Cyclin E during the very earliest period of G1 phase. Cell-cycle regulated Cyclin E is essential for maintenance of genome stability since cancer cells that have de-regulated Cyclin E fail to produce sufficient pre-RCs during G1 phase and as a consequence have problems in S phase and accumulate DNA damage (*Ekholm-Reed et al., 2004*; *Jones et al., 2013*; *Odajima et al., 2010*). Of interest is our observation that at the time CDC6 is recruited to the *CCNE1* promoter, ORC1 binding to the promoter declines.

In human cells, ORC1 shows a dynamic, temporally regulated nuclear localization pattern such that in early G1 phase it is distributed in a punctate pattern throughout the nucleus, but in late G1 phase, ORC1 predominantly binds to regions of the genome that replicate late in the subsequent S phase (*Kara et al., 2015*). We suggest that the events that occur at the *CCNE1* promoter, ORC1 binding first and recruitment of SUV39H1 and RB, followed by recruitment of Cyclin E-CDK2 and CDC6, may occur at many ORC1 binding sites in the genome, even those sites that are destined to assemble the entire ORC protein and promote pre-RC formation. SUV39H1 binding to ORC1 may influence its temporally dynamic nuclear localization during G1 phase. Such a scenario may also explain observations that have implicated a role for RB in DNA replication (*Bosco et al., 2001*; *Kennedy et al., 2000*; *Sterner et al., 1998*). The ORC1-CDC6 switch might temporally influence pre-RC assembly just like it does for *CCNE1* gene regulation, a possibility we are investigating.

## Materials and methods

### Cell culture and cell synchronization, with siRNA treatment or transient transfection of expression plasmids

U2OS, HEK293 and MCF7 cells were obtained from the Cold Spring Harbor Laboratory cell culture collection and cultured in DMEM containing high glucose (Gibco) supplemented with 10%

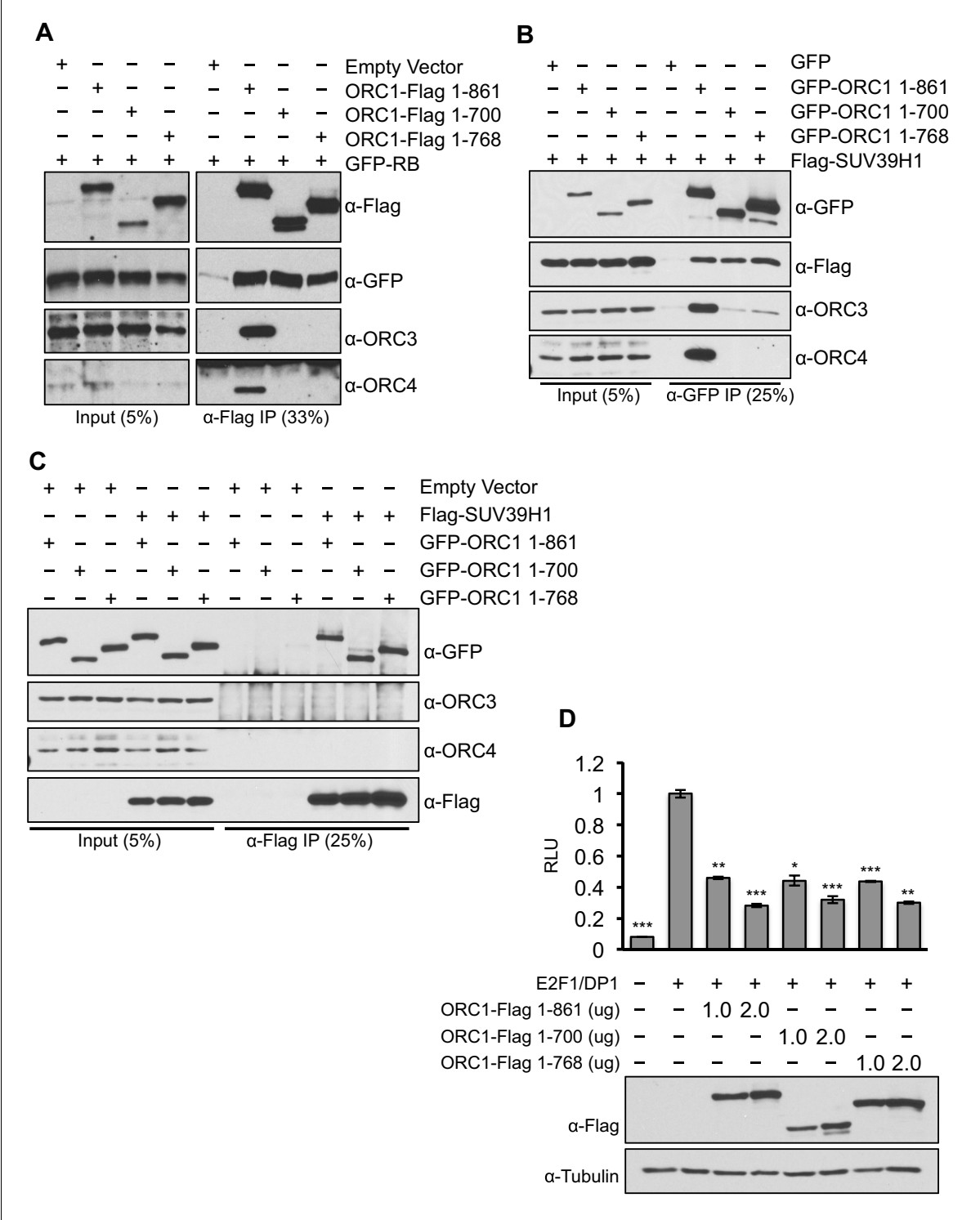

**Figure 7.** ORC1 mutants separate its role as a transcription co-repressor from its role in DNA replication. (**A**) HEK293 cells were transfected with ORC1-Flag or its truncation mutants and GFP-RB as indicated. Whole cell extracts were immunoprecipitated with anti-Flag antibody and immunoprecipitates were analyzed by immuoblot with the indicated antibodies. (**B** and **C**) In vivo interaction between SUV39H1 and ORC1 or its truncation mutants. GFP-tagged wild-type ORC1 (1–861) or its truncation mutants (1–700 and 1–768) plasmids were co-transfected into HEK293 cells with Flag-SUV39H1 plasmid and either GFP-vector or empty vector as a control plasmids. Immunoprecipitation with anti-GFP antibody (**B**) or anti-Flag antibody (**C**) from cell lysates of HEK293 cells expressing the indicated constructs, followed by immunoblotting with the indicated antibodies. (**D**). U2OS cells transiently transfected for 24 hr with increasing amounts of wild type ORC1-Flag (1–861) or truncation mutants (1–700 and 1–768). Relative luciferase activity normalized to co-

*Figure 7 continued on next page*

*Figure 7 continued*

transfected lacZ control. Experiments were in triplicate. Expression of proteins determined by Immunoblot; α-Tubulin as loading control. Statistical analysis was performed using the Student's t test. *p<0.005; **p<0.001; ***p<0.0005. ORC, Origin Recognition Complex.

inactivated fetal calf serum and Penicillin/Streptomycin. RB defective SaOS-2 cells were obtained from ATCC (HTB-85) and were cultured in McCoy's media supplemented with 15% inactivated fetal calf serum and Penicillin/Streptomycin. All the cell lines were tested negative for the mycoplasma contamination. To synchronize U2OS cells at G2/M boundary, 100 ng/mL of nocodazole was added to fresh medium for 16 hr. After 16 hr of block the cells were washed two times with 1x Phosphate Buffered Saline (PBS) and subsequently, released into the fresh media. For depletion of ORC1 protein in synchronized U2OS cells, the U2OS cells transiently transfected with 100nM of siRNAs (control GFP as well as ORC1 siRNA) using Lipofectamine RNAiMax and at the same time were treated with nocodazole, then the cells were incubated and released as described. Plasmids expressing exogenous genes were transfected using 2.5 µg of DNA except where indicated using lipofectamine 2000 transfection reagents (ThermoFisher Scientific; Waltham, MA). The sequences of the siRNAs used are listed in the *Supplementary file 1*.

## Plasmid construction and mutagenesis

Plasmids expressing GFP-RB, 10–4 *CCNE1* (Addgene: Cyclin E gene) promoter, E2F1, DP1, HDAC1 and pGL2-GAL4-UAS-Luc were purchased from Addgene. pCMV-LacZ plasmid was purchased from Clontech. Plasmid Flag-SUV39H1 was a gift from Peter Zhou (*Dong et al., 2013*). ORC1-Flag or its

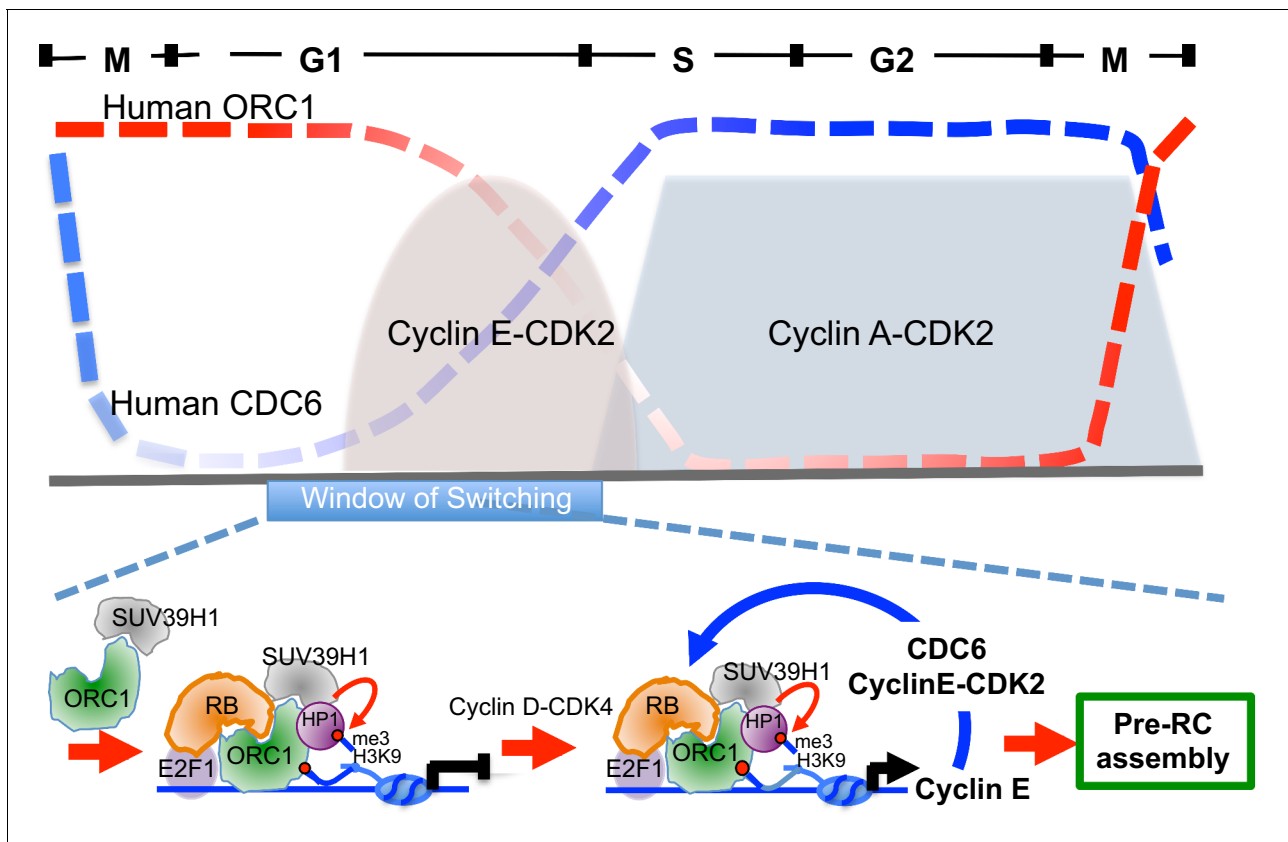

**Figure 8.** Model showing contrasting roles of DNA replication proteins ORC1 and CDC6 in the regulation of *CCNE1* transcription and commitment to pre-RC assembly and cell division. Top panel summarizes the cycle levels of ORC1 and CDC6. The bottom panel shown the ORC1 associated complexes at different stages of the cell division cycle. Blue arrow, positive feedback inhibition of ORC1-RB interaction.

mutant derivatives, ORC3-Flag, ORC4-Flag and T7-SUV39H1 plasmids were generated with a C-terminal Flag tag or N-terminal T7 tag in the pLPC vector (*McCurrach et al., 1997*) (gift from Scott Lowe, Memorial Sloan Kettering Cancer Center) for mammalian expression. Human ORC1 or its mutants, RB or its mutants, CDC6 or its mutants and SUV39H1 or its mutants were also cloned into pEGFP-C1 (Clontech, Mountain View, CA) for transient overexpression in mammalian cells. GAL4-DBD, GAL4-ORC1 and GAL-SUV39H1 plasmids were made by replacing the GFP insert in pEGFP-C1 constructs with in frame GAL4 DBD sequences. MBP-GFP-ORC1 plasmid is made by cloning GFP-ORC1 insert (amplified from GFP-ORC1 plasmid) in pMALc2E vector, while MBP-RB plasmid is made similar to MBP-ORC1 as described previously (*Hossain and Stillman, 2012*) for protein expression in bacterial cells. Human RB or its mutants, CDC6 or its mutant, SUV39H1 or its mutant, HP1α and HDAC1 were also cloned in pGEX-6P1 vector to produce Glutathione S transferase (GST) fusion proteins in bacterial cells. The mutant plasmids were generated following the site directed mutagenesis protocol (Quickchange, Stratagene, CA). All the plasmid constructs were verified by sequencing. The oligonucleotide sequences used to generate the plasmids are listed in the *Supplementary file 1*.

## Recombinant protein expression and purification

Wildtype ORC1, GFP-ORC1 and RB were fused to the maltose binding protein (MBP) in frame by cloning in the bacterial expression vector pMALc2E (New England Biolab). To generate Glutathione S transferase (GST) fusion proteins, wild type RB or its mutants (R661W and N757F), HDAC1, SUV39H1 or its fragments, HP1α or CDC6 were cloned into the bacterial expression vector pGEX6P1 (GE Healthcare Life Sciences, NJ). The MBP fusion recombinant proteins were expressed and purified using amylose beads according to the procedure described previously (*Hossain and Stillman, 2012*). For GST fusion proteins, transformed E. coli BL21 cells with their respective plasmids were induced for 12 hr with 0.3 mM of IPTG at 16°C. The induced cells were pelleted, washed, and further lysed with sonication in a lysis buffer A containing 25 mM Tris-HCl at pH 7.5, 150 mM NaCl, 0.02% NP-40, 5 mM benzamidine-HCl, 1 mM phenylmethylsulfonylfluoride, Protease cocktail inhibitor tablets [Roche], 10% glycerol) plus 100 mg/ml lysozyme. The lysed bacterial cells were centrifuged and the clarified supernatant is incubated with pre-washed Glutathione sepharose beads for 3 hr at 4°C. The bead-bound proteins were washed with three column volumes of buffer A plus 0.05% NP-40 + 500 mM NaCl and further with five column volumes of buffer A alone. Fusion protein was eluted in a stepwise manner with buffer A containing 20 mM reduced glutathione, pH7.5. Fractions containing purified proteins were pooled, concentrated, and dialyzed, and protein concentration was estimated using a standard Bradford protein assay.

## Pull down assays

For MBP-RB or MBP-ORC1 pull down assay, the same protocol as described previously was used (*Hossain and Stillman, 2012*), except that here we have used a different binding buffer with following composition; 25 mM Tris-Cl at pH 7.5, 100 mM KCl, 0.1% Nonidet P-40, 0.1 mM EDTA, 5 mM magnesium acetate, 1 mM DTT. For MBP-ORC1 pull down, Cyclin E-CDK2 protein was incubated with wild type GST-RB or its mutants in the presence or absence of 1 mM ATP. The pull down was further immunoblotted with anti-GST antibody (27-4577-01; GE Healthcare Life Sciences, NJ).

The histone peptide pull down assay followed the procedure described previously (*Hossain and Stillman, 2012*) with the minor modification of binding buffer composition (50 mM Tris-HCl at pH 7.5, 150 mM NaCl, 0.05% NP-40). The pull down was silver stained or immunoblotted with anti-MBP antibody (E8038S; New England Biolab). The biotin-labeled histone H3 peptides were purchased from AnaSpec (Fremont, CA) and bound to Streptavidin beads (Sigma-Aldrich, St. Louis, MO) prior to pull down studies.

For GST pull down assay, bead-bound GST fusion proteins (RB, HDAC1, SUV39H1 or its mutants, HP1α, and GST-CDC6 or its mutant) were incubated with either MBP-ORC1 or MBP-RB or Cyclin E-CDK2 protein. The composition of binding buffer used in the assay was; 25 mM Tris-Cl at pH 7.5, 150 mM KCl, 0.15% Nonidet P-40, 0.1 mM EDTA, 5 mM magnesium acetate, 1 mM DTT. The pull down was further immunoblotted with respective anti-GST (27-4577-01; GE Healthcare Life Sciences, NJ) or anti-Cyclin E antibodies (sc-247; Santa Cruz Biotechnology, Dallas, TX).

For titration-based binding experiments, increasing molar amounts of GST-CDC6 and/or Cyclin E-CDK2 along together with equimolar amounts of MBP-RB (20nM) and MBP-GFP-ORC1 (20nM) were used in different combinations. The reaction mixture was incubated for 4 hr at 4°C followed by 1 hr more incubation with bead coupled anti-GFP antibody (ABP-NAB-GFPA025; Allele Biotech, San Diego, CA) to precipitate MBP-GFP-ORC1. The reaction mixture was incubated and washed with binding buffer with following composition; 25 mM Tris-Cl at pH 7.5, 150 mM KCl, 0.15% Nonidet P-40, 0.1 mM EDTA, 5 mM magnesium acetate, 1 mM DTT and 1 mM ATP. The pull down was immunoblotted with the following antibodies; anti-ORC1 antibody (pKS1-40), anti-RB antibody (#9309; Cell Signaling, Danvers, MA), anti-GST (27-4577-01; GE Healthcare Life Sciences, NJ) and anti-Cyclin E antibody (sc-247; Santa Cruz Biotechnology, Dallas, TX).

## Immunoprecipitation

For expression of proteins, HEK293 cells were transiently transfected with the indicated plasmids with lipofectamine 2000 transfection reagents (ThermoFisher Scientific). Immunoprecipitation from HEK293 cells was performed using the procedure described previously (*Hemerly et al., 2009*) with slight modification in the protocol. Following expression of proteins, the cells were harvested and washed in PBS and lysed in a buffer containing 20 mM Tris-HCl pH7.5, 200 mM NaCl, 0.3% NP-40, 5 mM $MgCl_2$, 0.1 mM EDTA, 10% Glycerol, 1 mM DTT, 1 mM $CaCl_2$, 20 uM MG132 and protease as well as phosphatase inhibitor tablets (Roche). Benzonase (Sigma-Aldrich, St. Louis, MO) was added to the buffer and the suspension incubated for 30 min on ice with intermittent mixing. The concentration of NaCl and NP-40 was reduced to 100 mM and 0.15%, respectively with dilution buffer after 30 min incubation on ice. The extract was centrifuged at 14, 000 rpm for 15 min at 4°C. The proteins were precipitated with specific antibodies as indicated in figure legends using FLAG, GFP, ORC2 or ORC3 antibodies. The whole cell extract was first incubated with antibodies for 4 hr and subsequently, 2 hr with pre-washed gamma bind G sepharose beads with end-to end shaking at 4°C. The beads were washed 3 times with washing buffer containing 20 mM Tris-HCl pH7.5, 100 mM NaCl, 0.1% NP-40, 5 mM $MgCl_2$, 0.1 mM EDTA, 10% Glycerol, 1 mM DTT and protease as well as phosphatase inhibitor tablets from Roche. Finally, the washed beads were suspended in Laemmli sample buffer and 8% SDS-PAGE gels were run and immunoblotted. For immunoprecipitation of endogenous ORC1 and SUV39H1 proteins from U2OS or MCF7 cells and SaOS-2 cells, we have used monoclonal ORC1 antibody coupled to magnetic beads as described previously (*Kara et al., 2015*) as well as rabbit polyclonal SUV39H1 antibody (A302-128A; Bethyl Laboratories), respectively.

For immunoprecipitations, the following antibodies were used: rabbit polyclonal anti-Flag antibody (F7425; Sigma), GFP nAb (ABP-NAB-GFPA025; Allele Biotech, San Diego, CA), polyclonal anti-SUV39H1 antibody (A302-128A; Bethyl Laboratories), mouse monoclonal ORC1 78-1-172 (*Kara et al., 2015*), rabbit polyclonal anti-ORC2 antibody (CS205-5) (*Prasanth et al., 2004*), and rabbit polyclonal anti-ORC3 (CS1890) antibody (*Siddiqui and Stillman, 2007*). For immunoblots, monoclonal FLAG antibody (F1804; Sigma), polyclonal GFP antibody (G1544; Sigma), monoclonal mouse SUV39H1 antibody (05–615; Millipore), monoclonal mouse T7 antibody (Cold Spring Harbor Laboratory antibody facility), mouse monoclonal anti-ORC1 antibody (pKS1-40) (*Hemerly et al., 2009*), monoclonal mouse ORC2 antibody (920-2-44) (*Siddiqui and Stillman, 2007*), mouse monoclonal ORC3 antibody (PKS1-16) (*Prasanth et al., 2004*), goat polyclonal anti-ORC4 antibody (ab9641; Abcam, Cambridge, MA), monoclonal mouse E2F-1 antibody (KH95; Santa Cruz Biotechnology, Dallas, TX) and rabbit GAL4 antibody (sc-577; Santa Cruz Biotechnology, Dallas, TX) were used.

## RNA extraction and qRT-PCR

Nocodazole arrested U2OS cells were transfected with 100 nM siRNA (Dharmacon Inc., Lafayette, CO), and RNA was prepared at the indicated time points post-drug release using the RNeasy Mini Kit (Qiagen cat. #74104) including on-column DNase digestion (Qiagen cat. # 79254) and eluted in the supplied RNase-free water. The cDNA used for Q-PCR was prepared from 1µg each RNA sample using TaqMan Reverse Transcription Reagents (Applied Biosystems #N808-0234) with random hexamer priming in a GeneAmp PCR system 9700 thermocycler. Each Q-PCR reaction was prepared using 2 µL of 1-to-20 diluted cDNA and 13 µL LightCycler 480 SYBR Green I Master Mix (Roche #04887352001) and were performed in 384-well plates using the LightCycler 480 (Roche) as per manufacturer's instructions.

For semi-quantitative RT-PCR, equal amounts of RNA (0.2 µg) were used for RT-PCR using the Qiagen's One-Step RT PCR kit following manufacturer's instructions. PCR was performed for 22 cycles and subsequently, run on 1.8% agarose gel. The primer sequences used for RT-PCR analysis are listed in the *Supplementary file 1*.

## Luciferase reporter assay

For the luciferase reporter assay, 0.3 x 10$^6$ U2OS cells were seeded in six-well plates. Cells were transfected with 0.5 µg *CCNE1* promoter luciferase plasmid (p10-4), 50 ng E2F1 plasmid, 50 ng of DP1 plasmid and 20 ng LacZ plasmid. Together with above plasmids, the cells were also transfected either with ORC1-Flag or its mutants, ORC3-Flag, ORC4-Flag, T7-SUV39H1, GFP-SUV39H1 or its mutant, GFP-CDC6 or its mutant for overexpression at the indicated amounts in micrograms. The cells were transfected using Lipofectamine 2000 (Life Technologies) for 24 hr. For GAL4-based luciferase assay, the U2OS cells were transfected with 0.5 µg of GAL4-UAS-Luciferase promoter plasmid and 20 ng of LacZ plasmid along with GAL4-DBD fused ORC1 or SUV39H1 at the indicated amounts. In each of the experiments, the amount of plasmid was kept constant by the addition of empty vector DNA. For siRNA-mediated depletion of proteins and luciferase measurement, 100 nM of specific siRNAs targeting ORC1, SUV39H1 or CDC6 were used together with p10-4, E2F1/DP1 and LacZ plasmids. GFP siRNA is used as negative control. β-Galactosidase activity was measured as described previously (*Smale, 2010*), while luciferase activity was measured using luciferase (Promega) luminescent assay kit according to the manufacturer's instructions. Luciferase activities were normalized to β-galactosidase activities and denoted as relative light units (RLU). Expression or depletion of proteins was confirmed by Western blot. All the experiments were done in triplicates.

## Chromatin immunoprecipitation

MCF7 and U2OS cells were used for chromatin immunoprecipitation. The cells were harvested by trypsinization and washed with cold PBS. The cells (3.0x10$^7$) were fixed with 1% formaldehyde for 10 min at room temperature. The cross-linking was stopped by addition 125 mM of glycine for 5 min on ice. The fixed cells were washed with cold PBS and lysed for 10 min on ice with buffer containing 10 mM Tris pH 8.0, 10 mM NaCl, 2 mM MgCl$_2$, 0.4% NP40, 1 mM DTT, 10% Glycerol, protease and phosphatase inhibitors. Preliminary experiments demonstrated that ORC1 was most efficiently extracted from the resulting cells by digestion of the cross-linked chromatin with micrococcal nuclease (MNase; Sigma) and extraction with high salt (300–600 mM NaCl). Thus, the nuclei were treated with MNase with 1 mM CaCl2 at 37°C (so that most of the extracted DNA ran on an agarose gel as ~1–4 nucleosome length), the reaction was quenched with 2 mM EGTA and washed with buffer containing 10 mM Tris pH 8.0, 10 mM NaCl, 2 mM MgCl$_2$, 1 mM DTT, 10% Glycerol, protease and phosphatase inhibitors. The pelleted residual nuclei were further lysed in buffer containing 20 mM Tris pH 8.0, 1 mM EDTA, 0.5 mM EGTA, 300 mM NaCl, 0.5% TritonX-100, 0.05% Sodium Deoxycholate, 0.1% IGE-PAL, 1 mM PMSF and protease/phosphatase inhibitors for 30 min at 4°C with rotation. The lysate was further diluted to bring the salt concentration to 200 mM and sonicated very briefly. The lysate was centrifuged (20,800 X g in an Eppendorf centrifuge) at 4°C to pellet down the debris and supernatant was used for pre-clearing with protein G Dynabeads (Invitrogen) for 2 hr 4°C. The antibodies were bound to protein G Dynabeads for 5 hr at 4°C in PBS containing 0.5% BSA. The antibodies used for chromatin immunoprecipitation were as follows: mouse monoclonal ORC1 [78-1-172; (*Kara et al., 2015*)], mouse RB (#9309, Cell Signaling), rabbit polyclonal SUV39H1 (A302-128A; Bethyl Laboratories), rabbit polyclonal CDC6 (CS1881), rabbit polyclonal histone H3K9me3 (ab8898, Abcam) as well as control mouse and rabbit IgG (Invitrogen). Pre-cleared nuclear extracts was incubated with antibody-bound beads overnight with rotation at 4°C. 0.4 ml of the nuclear extract was used for each IP and 0.1 ml was kept aside for 'Input' in Q-PCR analysis. Beads were then washed with three times with low-salt buffer (20 mM Tris pH8.0, 200 mM NaCl, 0.5% TritonX-100, 0.05% Sodium Deoxycholate, 0.1% IGE-PAL, 1 mM PMSF), three times high-salt buffer (20 mM Tris pH8.0, 500 mM NaCl, 0.5% TritonX-100, 0.05% Sodium Deoxycholate, 0.1% IGE-PAL, 1 mM PMSF), three times with lithium chloride buffer (20 mM Tris pH8.0, 200 mM NaCl, 250 mM LiCl, 0.5% TritonX-100, 0.05% Sodium Deoxycholate, 0.1% IGE-PAL, 1 mM PMSF) and twice with TE buffer (10 mM Tris pH8.0, 1 mM EDTA). Chromatin was eluted from beads twice with 100 ul elution buffer (100 mM sodium bicarbonate, 1% SDS) at room temperature. Protein-DNA crosslinks in the IP

samples as well as in input samples were reversed overnight by addition of 300 mM NaCl and 2 ug RNase A at 65°C. The ChIP and input samples were then incubated with 60 ug of proteinase K for 2 hr at 42°C. The samples were further extracted with phenol:chloroform:isoamyl alchol [IAA] (once for ChIP and twice for input) and once with chloroform extraction. The ethanol precipitation were done by adding 10 ug of glycogen, washed with 70% ethanol, air dried and then re-suspended in 50 ul of RNase-DNase free water. The purified DNA was used as template in different PCR amplifications (Applied Biosystems Thermocycler). The sequences of the various primers is listed in *Supplementary file 1*. Due to very high GC content of human *CCNE1* promoter, the number of PCR cycles was extensively and independently optimized with different DNA polymerases (*Supplementary file 2*) for each primer set to maintain linear amplification in all experiments. PCR products were resolved by 2% agarose gel electrophoresis. Images of ethidium bromide-stained DNAs were acquired using an UV trans-illuminator equipped with a digital camera. Intensities of the amplified PCR bands were quantitated by ImageJ software.

## Data analysis

Data are shown as the average ± the standard deviation (SD) of results of at least three independent experiments. In the luciferase assay, the statistical differences between cells overexpressing ORC1 and/or SUV39H1 or CDC6 or no E2F1/DP1 were compared to E2F1/DP1 alone without overexpression, and statistically evaluated by Student's t test analysis. The lines above the bar graph were used to indicate the statistical differences between overexpressing samples. In the luciferase assay with depletion of specific proteins, the statistical differences between siRNAs against the ORC1, SUV39H1 or CDC6 were compared to control GFP siRNA using the Student's t test.

## Acknowledgements

We thank Patty Wendel for technical assistance and Jaclyn Jansen for commenting on the manuscript. This research was supported by a grant from the US National Cancer Institute (CA13106) and by a NCI Cancer Center Grant (CA45508).

## Additional information

### Funding

| Funder | Grant reference number | Author |
| --- | --- | --- |
| National Cancer Institute | CA13106 | Bruce Stillman |
| National Cancer Institute | CA45508 | Bruce Stillman |

The funders had no role in study design, data collection and interpretation, or the decision to submit the work for publication.

### Author contributions

MH, Performed the experiments, Conception and design, Acquisition of data, Analysis and interpretation of data, Wrote the paper; BS, Conception and design, Analysis and interpretation of data, Wrote the paper

### Author ORCIDs

Manzar Hossain, http://orcid.org/0000-0003-3399-581X
Bruce Stillman, http://orcid.org/0000-0002-9453-4091

## Additional files

### Supplementary files

• Supplementary file 1. Oligonucleotides employed for this research.

• Supplementary file 2. DNA polymerases used for each primer pair in ChIP analysis.

## Major datasets

The following previously published dataset was used:

| Author(s) | Year | Dataset title | Dataset URL | Database, license, and accessibility information |
| --- | --- | --- | --- | --- |
| Dellino GI, Cittaro D, Piccioni R, Luzi L, Banfi S, Segalla S, Cesaroni M, Giacca M, Pelicci PG | 2013 | ORC1 ChIP | http://www.ncbi.nlm.nih.gov/geo/query/acc.cgi?acc=GSE37583 | Publicly available at Gene Expression Omnibus (accession no. GSE37583) |

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
