## [Decision Letter]

Thank you for submitting your work entitled "Opposing roles for ORC1 and CDC6 in an RB-SUV39H1 dependent switch for control of Cyclin E gene transcription" for consideration by *eLife*. Your article has been favorably evaluated by Jessica Tyler (Senior editor) and three reviewers, one of whom, Michael Botchan, is a member of our Board of Reviewing Editors.

The reviewers have discussed the reviews with one another and the Reviewing Editor has drafted this decision to help you prepare a revised submission.

This manuscript builds on a deep vein pursued for many years in the Stillman laboratory focused on the functions of the ORC-1 protein in human cells. As the central ATPase and the major DNA binding subunit of the ORC complex, ORC1 is of course key for the initiation of DNA replication; the protein also functions in the regulation of centrosomes and centrioles in direct ways through its negative regulation of Cyclin E-dependent CDK activity. The present story ostensibly is about the roles of ORC1 in negative regulation of the transcription of the Cyclin E gene (*CCNE1*) through the recruitment of SUV39H1, binding directly to the proximal regulatory regions of the gene. The model put forward is that together with Rb, E2F1/DP and HP1 the transcription is turned down. The work then builds upon the finding that Cdc6 interferes in many of their assays with this negative regulation. All together a fascinating and potentially exciting proposal is made that links all of this into a model for how an autocatalytic stimulatory cascade, starting with cyclin D/Cdk activity in early G1 might feed into a control for a switch to enough Cyclin E expression to execute progression through the restriction point. We believe that the work is at present not yet ready for a publication in *eLife* but would be with further critical new studies that substantiate the overarching and exciting new ideas.

Summary:

The work does present many experiments that support the conclusions and certainly the biochemical studies establish many of the protein/protein interactions critical for the models. Si RNA to ORC1 does lead to enhanced Cyclin E gene mRNA and *CCNE1* reporter constructs transfected with ORC1 or mutants of ORC1 respond as predicted and are antagonized by CDC6 expression. Many other such experiments are consistent with the central proposals as are the scholarly arguments tying together points from the literature. However, one finds most of the cell-based findings dependent upon over-expression of recombinant proteins and acute transfection protocols.

Essential revisions:

1) Depletion of ORC1 protein does lead to more mRNA for *CCNE1* but might that result be indirect? ORC1 can bind to many proteins, including the targets defined here. Therefore it is essential for this work to show directly that, under normal progression, ORC1 binds to the endogenous gene (where?), along with Rb, the methylase – or HP1 and E2F1, and that, upon progression, the CDC6 protein actually replaces the repressive components with a loss of local Rb.

2) There are no chromatin binding experiments (ChIP), which would be required to make definitive conclusions. In particular, the authors would need to show using ChIP that ORC1, RB, and Suv39h1 are recruited at the appropriate stage of the cell cycle and that the result of such binding is transcriptional repression. This comment from reviewer #2 overlaps with #1 but adds the following: functional, gene ablation experiments should also be performed, again, by studying physiological levels of each protein without protein over-production.

3) From reviewer #3: The paper has many luciferase reporter assays to show the repression of a transiently transfected CCNE promoter by over-expressed Rb+ORC1+SUV39H1. What is missing is a clear demonstration that (a) the endogenous CCNE promoter is epigenetically repressed by the recruitment of this ORC1-dependent complex under normal levels of expression of E2F1/DP1, Rb, ORC1 and SUV39H1 and (b) that the physical interaction of endogenous levels of ORC1 with Rb and SUV39H1 is essential for this regulation. If this deficit is rectified, there could be two papers: "ORC1 inhibits CCNE by epigenetic modification of the CCNE promoter" and "CDC6 together with Cyclin E-CDK2, disrupts an ORC1-RB-SUV39H1 repressor complex to activate the CCNE promoter". While we do not suggest writing two papers the authors should in a revision more completely describe how CDC6 actually works for activation (see below #9).

4) siORC1 increases cyclin E mRNA 1.3-1.4X. This is the result that put the investigators on the way to this interesting hypothesis. However, the hypothesis makes some clear predictions that have not been tested: does siORC1 decrease H3K9me3 marks on the CCNE promoter? Does siORC1 decrease the association of Rb and of SUV39H1 at the CCNE promoter?

5) If ORC1 represses the CCNE promoter in early G1 by facilitating the Rb-SUV39H1 complex, another important test will be to show that ORC1 ChIPs to the CCNE promoter, in an Rb dependent manner at the E2F1/DP1 ChIP sites. This experiment is of course another suggestion for testing a major conclusion and other approaches may suffice.

6) The interaction of ORC1 with RB or with SUV39H1 is at the center of this hypothetical repressor complex. To prove this hypothesis the authors need to find a well defined amino acid change or mutant in ORC1 that disrupts this interaction and demonstrate that such a mutant ORC1 fails to repress the cyclin E promoter in the reporter assays used. Even better, if an ORC1 resistant to Si RNA is available and used, then such a mutant ORC1 should fail to restore normal expression of CCNE when endogenous ORC1 has been knocked down by siRNA.

7) Since the ORC1-SUV39H1 interaction occurs in the absence of Rb, does siORC1 still de-repress CCNE mRNA in Rb- cells such as SaOS-2. In other words, is Rb required for the repression by ORC1 on the endogenous CCNE promoter?

8) 293 cells express adenovirus E1A, believed to inactivate Rb. Despite this, overexpressed of GFP-Rb interacts with ORC1. Is this sufficient to repress the endogenous CCNE promoter despite the presence of E1A? Will this increase H3K9me2 to the CCNE promoter?

9) Similar experiments are necessary to show that endogenous CDC6 de-represses the endogenous CCNE promoter by disrupting the endogenous Rb-ORC1-Suv39H1 complex, but as said above, this might require writing two papers if the experiments are difficult and time consuming.

---

## [Author Response]

*Essential revisions:*

1) Depletion of ORC1 protein does lead to more mRNA for CCNE1 but might that result be indirect? ORC1 can bind to many proteins, including the targets defined here. Therefore it is essential for this work to show directly that, under normal progression, ORC1 binds to the endogenous gene (where?), along with Rb, the methylase – or HP1 and E2F1, and that, upon progression, the CDC6 protein actually replaces the repressive components with a loss of local Rb.

We agree with the reviewers that it is essential to show the binding of ORC1 protein to *CCNE1* promoter. At the time of initial submission of this paper to *eLife*, we tried to do conventional chromatin immunoprecipitation (ChIP) assays for ORC1 protein but our attempts failed due to highly GC-rich region of *CCNE1* promoter. The region -363 to +63 of *CCNE1* promoter is 85% GC rich, where all of the E2F1 sites lie and RB protein binds. Thus conventional ChIP-seq experiments did not work. Because of the reviewers’ comments, we invested considerable time in designing specific primer pairs that would allow detection of protein binding using ChIP and PCR. We also optimized the immunoprecipitation protocol using a monoclonal anti-ORC1 antibody that we made that specifically immunoprecipitates human ORC1 and found that ChIP using this antibody best worked with chromatin fragmented by micrococcal nuclease, followed by salt extraction from nuclei (ORC1 is known to bind tightly with a nuclear structure). Using this new protocol and synchronization of cells into early G1 phase, we were able to detect binding of ORC1, SUV39H1 and RB to the *CCNE1* promoter in early G1 phase and transient CDC6 binding to the *CCNE1* promoter precisely at the time when the promoter is activated in vivo. In the revised manuscript, we provide the data for binding ORC1, RB, SUV39H1, H3K9me3 and CDC6 proteins to the *CCNE1* promoter in asynchronously growing MCF7 cells as well as in synchronized U2OS cells.

2) There are no chromatin binding experiments (ChIP), which would be required to make definitive conclusions. In particular, the authors would need to show using ChIP that ORC1, RB, and Suv39h1 are recruited at the appropriate stage of the cell cycle and that the result of such binding is transcriptional repression. This comment from reviewer #2 overlaps with #1 but adds the following: functional, gene ablation experiments should also be performed, again, by studying physiological levels of each protein without protein over-production.

We now provide the requested ChIP data for ORC1, RB, SUV39H1 and CDC6 proteins binding to *CCNE1* promoter from synchronized U2OS cells. The U2OS cells were arrested with nocodazole and released for different times (as indicated in manuscript) and the fragmented chromatin immunoprecipitated using ORC1, RB, SUV39H1 and CDC6 antibodies to determine both the location and time of binding to the *CCNE1* promoter. We have also performed functional gene ablation experiments by depletion of ORC1 in U2OS cells and investigated the binding of histone H3K9me3 and SUV39H1 to the *CCNE1* promoter. Our new results show that upon ORC1 depletion, the H3K9me3 mark is drastically reduced at the *CCNE1* promoter, as well as slight reduction in SUV39H1 binding.

3) From reviewer #3: The paper has many luciferase reporter assays to show the repression of a transiently transfected CCNE promoter by over-expressed Rb+ORC1+SUV39H1. What is missing is a clear demonstration that (a) the endogenous CCNE promoter is epigenetically repressed by the recruitment of this ORC1-dependent complex under normal levels of expression of E2F1/DP1, Rb, ORC1 and SUV39H1 and (b) that the physical interaction of endogenous levels of ORC1 with Rb and SUV39H1 is essential for this regulation. If this deficit is rectified, there could be two papers: "ORC1 inhibits CCNE by epigenetic modification of the CCNE promoter" and "CDC6 together with Cyclin E-CDK2, disrupts an ORC1-RB-SUV39H1 repressor complex to activate the CCNE promoter". While we do not suggest writing two papers the authors should in a revision more completely describe how CDC6 actually works for activation (see below #9).

We do not suggest that there is “epigenetic” repression because this would imply inheritance for multiple generations, but we suggest that this system is reset every cell cycle so that a decision can be made about expression of Cyclin E. We suggest that repression is caused by DNA binding proteins recruited to the *CCNE1* promoter and establishment of a histone H3K9me3 mark that can bind HP1. The combination of ORC1, RB and SUV39H1 results in the repressive histone H3K9me3 mark at the promoter. We provide new data that shows that this repressive mark on the endogenous promoter is dependent on ORC1. Furthermore, our new data from synchronized U2OS cells shows binding of ORC1, RB, SUV39H1 to the *CCNE1* promoter in early G1 phase and then at the time when Cyclin E levels increase, CDC6 is transiently recruited to the endogenous *CCNE1* promoter. These data are entirely consistent with our model that ORC1 and CDC6 have opposing roles in control of *CCNE1* gene transcription. The experiments were done under normal levels of expression of E2F1/DP1, RB, ORC1 and SUV39H1 proteins. The significance of CDC6 in transcriptional activation is more elaborated in the revised manuscript. Entirely consistent with the model, we showed that increased expression of CDC6 actually causes increases in endogenous Cyclin E protein levels very early in G1 phase. We prefer to keep all the data in a single paper, albeit it now has increased in size due to the new data.

4) siORC1 increases cyclin E mRNA 1.3-1.4X. This is the result that put the investigators on the way to this interesting hypothesis. However, the hypothesis makes some clear predictions that have not been tested: does siORC1 decrease H3K9me3 marks on the CCNE promoter? Does siORC1 decrease the association of Rb and of SUV39H1 at the CCNE promoter?

We have now performed ChIP with H3K9me3 and Suv39h1 antibodies on the *CCNE1* promoter following siRNA mediated depletion of endogenous Orc1 protein. There is a dramatic reduction in the repressive H3K9me3 mark at the promoter following ORC1 depletion. The results are presented in the revised manuscript and its significance discussed.

5) If ORC1 represses the CCNE promoter in early G1 by facilitating the Rb-SUV39H1 complex, another important test will be to show that ORC1 ChIPs to the CCNE promoter, in an Rb dependent manner at the E2F1/DP1 ChIP sites. This experiment is of course another suggestion for testing a major conclusion and other approaches may suffice.

In the original manuscript we did not emphasize that ORC1 is recruited in an RB dependent manner, but we have stressed that ORC1 protein binds more strongly to SUV39H1 and is responsible, in part, for recruiting SUV39H1 to the *CCNE1* promoter.

6) The interaction of ORC1 with RB or with SUV39H1 is at the center of this hypothetical repressor complex. To prove this hypothesis the authors need to find a well defined amino acid change or mutant in ORC1 that disrupts this interaction and demonstrate that such a mutant ORC1 fails to repress the cyclin E promoter in the reporter assays used. Even better, if an ORC1 resistant to Si RNA is available and used, then such a mutant ORC1 should fail to restore normal expression of CCNE when endogenous ORC1 has been knocked down by siRNA.

We have spent considerable time and effort mapping the interactions between RB and ORC1 on one hand and ORC1 and SUV39H1 on the other. We prefer not to put this data in the paper since it has proven to be really complicated and is not yet complete for publication. For example, we have the mapped the region in ORC1 required for binding the SUV39H1 protein using in vitro interactions, but when mutant versions were used in vivo, it was not as straight forward as the in vitro data. We suspect that this might be due to the fact that RB interacts also with SUV39H1. We also mapped the regions of ORC1 that interacts with RB and found that the N-terminus of ORC1 binds the C-terminus of RB, but in addition, the C-terminus of ORC1 binds the N-terminus of RB. Thus the interaction between ORC1 and RB involves two separate domains in both proteins. Thus, in the present study, it was difficult to map the minimal interaction region in ORC1 required to bind either SUV39H1 or RB proteins and do the experiments suggested. We are continuing to study these protein-protein interactions, but think the new additional data strengthens our paper without this protein interaction data.

7) Since the ORC1-SUV39H1 interaction occurs in the absence of Rb, does siORC1 still de-repress CCNE mRNA in Rb- cells such as SaOS-2. In other words, is Rb required for the repression by ORC1 on the endogenous CCNE promoter?

In order to address the effect of ORC1 depletion in RB negative SaOS-2 cells, we first compared protein levels of the two osteosarcoma cell lines, U2OS (RB^+/+^) and SaOS-2 (RB^-/-^). To our surprise, compared to U2OS cells, the level of cyclin E was extremely low in the RB negative SaOS-2 cells (see Figure 9), while we detect comparable levels of ORC1, SUV39H1, Cyclin A, ORC3 and tubulin proteins in both cell lines. This data shows that apart from RB, other proteins in SaOS-2 cells must control the level of cyclin E or that the Cyclin E gene promoter is altered by mutation. Furthermore, upon ORC1 depletion, the *CCNE1* mRNA increased slightly by 48hr in asynchronously growing SaOS-2 cells. We chose not to include this data in the revised manuscript because it looks as if the SaOS2 cells have very low levels of Cyclin E independent of RB and figuring this out will be a distraction from the main point of our paper.

Author response image 1.Protein levels in U2OS cells and SaOS-2 cells detected by immunoblot.**DOI:**
http://dx.doi.org/10.7554/eLife.12785.021

8) 293 cells express adenovirus E1A, believed to inactivate Rb. Despite this, overexpressed of GFP-Rb interacts with ORC1. Is this sufficient to repress the endogenous CCNE promoter despite the presence of E1A? Will this increase H3K9me2 to the CCNE promoter?

We have not used the 293 cells for *CCNE1* promoter assay; the 293 cell line was only used to detect the interactions of over-expressed proteins.

*9) Similar experiments are necessary to show that endogenous CDC6 de-represses the endogenous CCNE promoter by disrupting the endogenous Rb-ORC1-Suv39H1 complex, but as said above, this might require writing two papers if the experiments are difficult and time consuming.*

We have shown in the manuscript that co-operation of CDC6/Cyclin E-CDK2 is required to reduce the interaction ORC1 protein to RB protein and it is already known that phosphorylation of RB causes its dissociation E2F1 protein. There is a paper published in 2014 by Park et al. (PMID: 24728993), wherein they have shown that phosphorylation of SUV39H1 by CyclinE-CDK2 (or CyclinA-CDK2) preferentially dissociates it from chromatin. Thus CDC6/Cyclin E-CDK2 cooperation might also lead to SUV39H1 phosphorylation and ultimately leading to falling apart of critical proteins involved in repressor complex. We have shown in our paper that CDC6 expression can activate the endogenous *CCNE1* promoter in very early G1 phase and that this activation requires CDC6 binding to Cyclin E-CDK2. This is entirely consistent with our model.